

# Environmental factors influencing cold-water coral ecosystems in the oxygen minimum zones on the Angolan and Namibian margins

Ulrike Hanz[1], Claudia Wienberg[2], Dierk Hebbeln[2], Gerard Duineveld[1], Marc Lavaleye[1], Katriina Juva[3], Wolf-Christian Dullo[3], André Freiwald[4], Leonardo Tamborrino[2], Gert-Jan Reichart[1,5], Sascha Flögel[3], Furu Mienis[1]

[1]NIOZ-Royal Netherlands Institute for Sea Research and Utrecht University, Department of Ocean Systems, Texel, 1797SH, Netherlands

[2]MARUM–Center for Marine Environmental Sciences, University of Bremen, Bremen, 28359, Germany

[3]GEOMAR Helmholtz Centre for Ocean Research, Kiel, 24148, Germany

[4]Department for Marine Research, Senckenberg Institute, Wilhelmshaven, 26382, Germany

[5]Faculty of Geosciences, Earth Sciences Department, Utrecht University, Utrecht, 3512JE, Netherlands

Correspondence to: Ulrike Hanz (ulrike.hanz@nioz.nl), +31222369466





## 23   Abstract

Fossil cold-water coral mounds overgrown by sponges and bryozoans were observed in anoxic
conditions on the Namibian margin, while mounds colonized by thriving cold-water coral reefs were
found in hypoxic conditions on the Angolan margin. These low oxygen conditions do not meet known
environmental ranges favoring cold-water corals and hence are expected to provide unsuitable habitats
for cold-water coral growth and therefore reef formation. To explain why the living fauna can
nevertheless thrive in both areas, present day environmental conditions at the southwestern African
margin were assessed. Downslope CTD transects and the deployment of bottom landers were used to
investigate spatial and temporal variations of environmental properties. Temporal measurements in the
mound areas recorded oscillating low dissolved oxygen concentrations of 0-0.17 ml l$^{-1}$ ($\triangleq$ 0-9 %
saturation) on the Namibian and 0.5-1.5 ml l$^{-1}$ ($\triangleq$ 7-18 % saturation) on the Angolan margin, which were
associated with relatively high temperatures (11.8-13.2°C and 6.4-12.6°C respectively). Semi-diurnal
barotrophic tides were found to interact with the margin topography producing internal waves with
excursions of up to 70 and 130 m for the Namibian and Angolan margins, respectively. These tidal
movements temporarily deliver water with more suitable characteristics to the coral mounds from
below and above the hypoxic zone. Concurrently, the delivery of high quantity and quality of suspended
particulate organic matter was observed, which serves as a food source for cold-water corals. On the
Namibian slope organic matter indicates a completely marine source and originates directly from the
surface productive zone, whereas on the Angolan margin the geochemical signature of organic material
suggest an additional mechanisms of food supply. A nepheloid layer observed above the cold-water
coral mound area on the Angolan margin may constitutes a reservoir of fresh organic matter, facilitating
a constant supply of food particles by tidal mixing. This suggests that the cold-water coral communities
as well as the associated fauna may compensate unfavorable conditions induced by low oxygen levels
and high temperatures with an enhanced availability of food. With the expected expansion of oxygen
minimum zones in the future due to anthropogenic activities, this study provides an example on how
ecosystems could cope with such extreme environmental conditions.




# 1. Introduction

Cold-water corals (CWCs) form 3D structures in the deep-sea, providing important habitats for dense aggregations of sessile and mobile organisms ranging from mega- to macrofauna (Henry and Roberts, 2007;van Soest et al., 2007) and fish (Costello et al., 2005). Consequently, CWC areas are considered as deep-sea hotspots of biomass and biodiversity (Buhl-Mortensen et al., 2010;Henry and Roberts, 2017). Moreover, they form hotspots for carbon cycling by transferring carbon from the water column towards associated benthic organisms (Oevelen et al., 2009;White et al., 2012). Some framework-forming scleractinian species, with *Lophelia pertusa* and *Madrepora oculata* being the most common species in the Atlantic Ocean (Freiwald et al., 2004;White et al., 2005;Roberts et al., 2006;Cairns, 2007), are capable of forming large elevated seabed structures, so called coral mounds (Wilson, 1979;Wienberg and Titschack, 2017;Titschack et al., 2015;De Haas et al., 2009). These coral mounds, consisting of coral debris and hemipelagic sediments, commonly reach heights between 20 and 100 m and can be several kilometers in diameter. They are widely distributed along the Atlantic margins, being mainly restricted to water depths between 200-1000 m, while records of single colonies of *L. pertusa* are reported from a broader depth range of 50-4000 m depth (Roberts et al., 2006;Hebbeln et al., 2014;Davies et al., 2008;Mortensen et al., 2001;Freiwald et al., 2004;Freiwald, 2002;Grasmueck et al., 2006;Wheeler et al., 2007).

A global ecological-niche factor analysis by Davies et al. (2008) and Davies and Guinotte (2011), predicting suitable habitats for *L. pertusa,* showed that this species generally thrives in areas which are nutrient-rich, well oxygenated and affected by relatively strong bottom water currents. Other factors potentially important for proliferation of *L. pertusa* include chemical and physical properties of the ambient water masses (aragonite saturation state, salinity, temperature) and water depth (Davies et al., 2008;Dullo et al., 2008;Flögel et al., 2014;Davies and Guinotte, 2011). *L. pertusa* is most commonly found at temperatures between 4-12°C and a very wide salinity range between 32 and 38.8 (Freiwald et al., 2004). Although they occur in such a wide range of temperature and salinity, the link of *L. pertusa* to particular salinity and temperature within the NE Atlantic led Dullo et al. (2008) to suggest that they are restricted to a specific density envelope of sigma-theta ($\sigma\Theta$) = 27.35-27.65 kg m$^{-3}$. In addition, the majority of occurrences of live *L. pertusa* come from sites with dissolved oxygen concentrations ($DO_{conc}$) between 6-6.5 ml l$^{-1}$ (Davies et al., 2008), with lowest recorded oxygen values being 2.1-3.2 ml l$^{-1}$ at CWC sites in the Gulf of Mexico (Davies et al., 2010;Schroeder, 2002;Brooke and Ross, 2014) or even as low as 1-1.5 ml l$^{-1}$ off Mauritania where CWC mounds are in a dormant stage showing only scarce living coral occurrences today (Wienberg et al., 2018;Ramos et al., 2017). Dissolved oxygen levels hence seem to



affect the formation of CWC structures as also shown by Holocene records obtained from the
Mediterranean Sea, which revealed periods of reef demise and growth in conjunction with hypoxia (with
2 ml l$^{-1}$ seemingly forming a threshold value for active coral growth; Fink et al., 2012).
Another essential constraint for CWC growth and therefore mound development in a generally food
deprived deep-sea is food supply. *L. pertusa* is an opportunistic feeder, exploiting a wide variety of
different food sources, including phytodetrius, phytoplankton, mesozooplankton, bacteria and dissolved
organic matter (Kiriakoulakis et al., 2005;Dodds et al., 2009;Gori et al., 2014;Mueller et al.,
2014;Duineveld et al., 2007). Transport of surface organic matter towards CWC sites at intermediate
depths has been found to involve either active swimming (zooplankton), passive sinking, advection, local
downwelling, and internal waves and associated mixing processes resulting from interactions with
topography (Davies et al., 2009;van Haren et al., 2014;Thiem et al., 2006;White et al., 2005;Mienis et al.,
2009;Frederiksen et al., 1992). Not only quantity but also quality of food particles is of crucial
importance for the uptake efficiency as well as ecosystem functioning of CWCs (Ruhl, 2008;Mueller et
al., 2014).
With worldwide efforts to map CWC communities, *L. pertusa* has also been found under conditions
which seem environmentally stressful or extreme in the sense of the global limits defined by Davies et
al. (2008) and Davies and Guinotte (2011). Examples are the warm and salty waters of the
Mediterranean and the high temperature variations along the US coast (Cape Lookout; Freiwald et al.,
2009;Mienis et al., 2014;Taviani et al., 2005). Environmental stress generally increases energy needs for
organisms to recover and maintain optimal functioning, which accordingly increases their food demand
(Sokolova et al., 2012). For the SW African margin one of the few records of living CWC come from the
Angolan margin (at 7°; Le Guilloux et al., 2009), which raises the question whether local environmental
factors limit CWC growth due to the presence of an Oxygen Minimum Zone (OMZ; see Karstensen et al.
2008), or data is lacking. Hydroacoustic campaigns nevertheless revealed extended areas off Angola and
Namibia with structures that morphologically resemble coral mounds structures known from the NE
Atlantic (M76-3, MSM20-1; Geissler et al., 2013;Zabel et al., 2012).
Two such mound areas on the margins off Namibia and Angola were visited during the RV *Meteor* cruise
M122 'ANNA' (ANgola/NAmibia) cruise in January 2016 (Hebbeln et al., 2017). During this cruise fossil
CWC mound structures were observed near Namibia, while flourishing CWC reef covered mound
structures were observed on the Angolan margin. The aim of the present study was to assess present-
day environmental conditions at the southwestern African margin that enable cold-water coral growth




in this low oxygen environment. To identify key parameters influencing CWC growth and therefore
mound development, hydrographic parameters as well as chemical properties of the water column were
measured, characterizing environmental conditions and food supply. These data are used to provide
new insights in susceptibility of CWCs towards extreme oxygen limited environments, in order to
improve understanding of the fate of CWC mounds in a changing ocean.

## 119   2. Material and Methods

### 120   *2.1 Setting*

#### 121   *2.1.1 Oceanographic setting*

The SW African margin is one of the four major eastern boundary regions in the world and is
characterized by upwelling of nutrient-rich cold waters (Shannon and Nelson, 1996). The availability of
nutrients triggers a high primary production, making it one of the most productive marine areas
worldwide with an estimated production of 0.37 Gt C/yr (Carr and Kearns, 2003). Remineralization of
high fluxes of organic particles settling through the water column result in severe mid-depth oxygen
depletion and an intense OMZ over large areas along the SW African margin (Chapman and Shannon,
1985). The extension of the OMZs is highly dynamic being controlled by upwelling intensity which
depends on the prevailing winds and two current systems along the SW African margin, i.e. the Benguela
and the Angola currents (Kostianoy and Lutjeharms, 1999;Chapman and Shannon, 1987; Fig. 1). The
Benguela Current originates from the South Atlantic Current, which mixes with water from the Indian
Ocean at the southern tip of Africa (Poole and Tomczak, 1999;Mohrholz et al., 2008;Rae, 2005) and
introduces relatively cold and oxygen-rich Eastern South Atlantic Central Water (ESACW; Poole and
Tomczak 1999) to the SW African margin (Mohrholz et al., 2014). The Angola Current originates from the
South Equatorial Counter Current and introduces warmer, nutrient-poor and less oxygenated South
Atlantic Central Water (SACW; Poole and Tomczak 1999) to the continental margin (Fig. 1a). While the
SACW flows along the continental margin the oxygen concentration is decreasing continuously due to
remineralisation processes on the SW African shelf (Mohrholz et al., 2008). Both currents converge at
around 14-16 °S, resulting in the Angola-Benguela Front (Lutjeharms and Stockton, 1987). In austral
summer, the Angola-Benguela Front can move southward to 23 °S (Shannon et al., 1986), thus
increasing the influence of the SACW along the Namibian coast (Junker et al., 2017;Chapman and
Shannon, 1987), contributing to the pronounced OMZ due to its low initial oxygen concentration (Poole
and Tomczak, 1999). ESACW is the dominant water mass at the Namibian margin during the main




upwelling season in austral winter, expanding from the oceanic zone about 350 km offshore, further in-
shore. (Mohrholz et al., 2014).

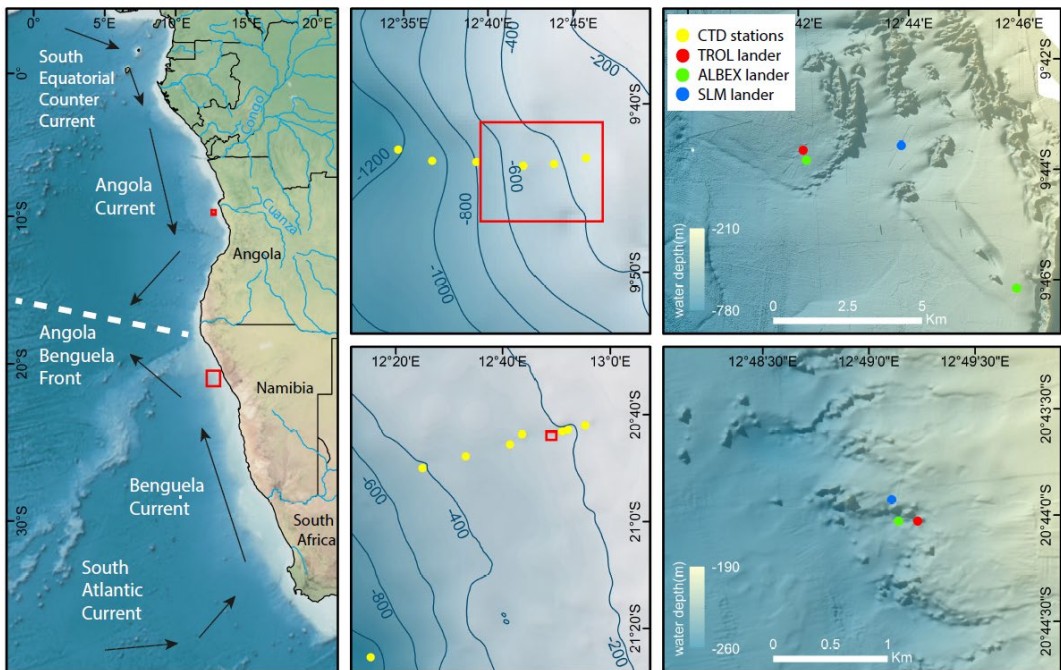

**Figure 1** (a) Overview map showing the research areas off Angola and Namibia (red squares) and main features of the surface
water circulation (arrows) and frontal zone (dashed line) in the SE Atlantic as well as the two main rivers discharging at the
Angolan margin. Detailed bathymetry maps of the Angolan (upper maps) and Namibian margins (lower maps) showing the
position of (b) CTD transects (note the deep CTD cast down to 1000 m water depth conducted off Namibia) and (c) bottom
lander deployments (red squares shown in (b) indicate the cutouts displayed in (c)).

### 2.1.2. Coral mound provinces along the Angolan and Namibian margins

During RV *Meteor* cruise M122 in 2016, over 2000 coral mounds were observed between 160-260 m
water depth on the Namibian shelf (Hebbeln et al., 2017). All mounds were densely covered with coral
rubble and dead coral framework (entirely consisting of *L. pertusa*), while no living corals were observed
in the study area (Hebbeln et al., 2017; Figs. 2a, b). Few species were locally very abundant, viz. a yellow
cheilostome bryozoan which was the most common species, and five sponge species. The bryozoans
were encrusting the coral rubble, whereas some sponge species reached heights of up to 30 cm (Fig. 2a,
b). The remaining CWC community consisted of an impoverished fauna overgrowing *L. pertusa* debris.
Commonly found sessile organism were actiniarians, zoanthids, hydroids, some thin encrusting sponges,
serpulids and sabellid polychaetes. The mobile fauna comprised asteroids, ophiuroids, two shrimp



species, amphipods, cumaceans and holothurians. Locally high abundances of *Suffogobius bibarbatus*, a
fish that is known to be adapted to hypoxic conditions, were observed in cavities underneath the coral
framework (Hebbeln, 2017). Corals collected from the surface of various Namibian mounds date back to
about 5 ka BP, pointing to a simultaneous demise of these mounds during the mid-Holocene
(Tamborrino et al., sumbitted).
On the Angolan margin CWC structures varied from individual mounds to long ridges. Some mounds
reached heights of more than 100 m above the seafloor. At shallow depths (~250 m) also some isolated
smaller mounds are present (Hebbeln et al. 2017). All mounds showed thriving CWC cover, which was
dominated by *L. pertusa* (estimated 99% relative abundance), *M. oculata* and solitary corals. Mounds
with a flourishing coral cover were mainly situated at water depths between 330-470 m, whereas single
colonies were found over an even broader depth range between 250-500 m (Figs. 2c, d; Hebbeln et al.,
2017). Additionally to CWCs, large aggregations of hexactinellid sponges (*Aphrocallistes, Sympagella*)
were observed. The scattered small mounds at shallower water depths were dominated by sponges and
only sparsely covered with living coral. In these areas, active suspension feeders, like sponges were most
commonly observed. First estimates for coral ages obtained from a gravity core records collected at one
of the Angolan coral mounds revealed continuous coral mound formation during the last 34 ka until
today (Wefing et al., 2017).





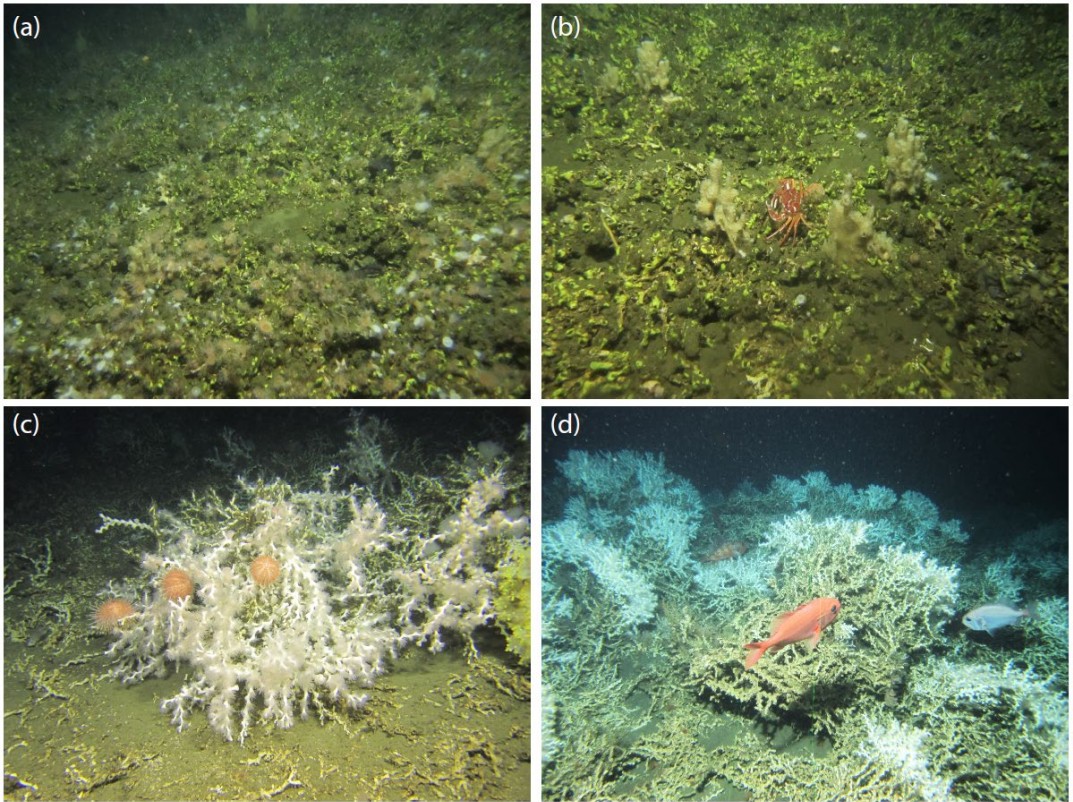


**Figure 2** ROV images (copyright MARUM ROV SQUID, Bremen, Germany) showing the surface coverage of cold-water coral
mounds discovered off Namibia (a, b) and Angola (c, d). Images were recorded and briefly described for their faunal
composition during RV *Meteor* cruise M122 "ANNA" (see Hebbeln et al. 2017). (a) Sylvester mound, 225 m water depth. Dead
coral framework entirely consisting of *Lophelia pertusa*. The framework is intensely colonized by the yellow bryozoan
*Metropriella* sp., zoanthids, actiniarians and sponges. Vagile fauna consists of asteroids and gobiid fishes (*Sufflogobius*
*bibarbatus*) that hide between hollows underneath the coral framework. (b) Sylvester mound, 238 m water depth. Dense coral
rubble (*L. pertusa*) heavily overgrown by *Metropriella* sp. and sponges. Note the decapod crab *Macropipus australis* (center of
the image). (c) Valentine mound, 238 m water depth. Live *Lophelia* colony being grazed by echinoids. Note the sponge
*Aphrocallistes* sp. with its actiniarian symbionts (right side of the image). (d) Buffalo mound, 345 m water depth. Living CWC
reef observed on top of an Angolan coral mound. Many fishes are present around the reef (*Helicolenus dactylopterus*,
*Gephyroberyx darwinii*).
## 2.2 Methodology
During RV *Meteor* expedition M122 in January 2016, two CTD transects and five short-term bottom
lander deployments (Table 1, Fig. 1) were carried out to measure near-bed environmental conditions



potentially influencing benthic habitats. In addition, weather data were continuously recorded by the RV
*Meteor* weather station, providing real-time information on local wind speed and wind direction.

### 196    *2.2.1 Lander deployments*

Sites for deployment of the NIOZ designed landers (ALBEX and TROL) were selected based on multibeam
bathymetric data. In each area, two landers were deployed simultaneously. On the Namibian margin
one lander was deployed on top of a mound structure (water depth 220 m), while the other lander was
deployed in close vicinity but off-mound (Fig. 1, Table 1). As both landers show similar trends for all
measured physical properties, only the dataset recorded by the ALBEX lander is presented here (see
Table 1). Off Angola, mounds with live corals were observed over a large depth zone (250-500 m). To
obtain as much information as possible over the entire mound zone, one lander (ALBEX) was first
deployed in the relatively shallow part of the mound zone at 340 m water depth, and after retrieval
redeployed in the deeper part of the zone at 530 m.  A second lander (TROL) was deployed at the
deeper part of the zone (530 m) for the full time period (Fig. 1, Table 1). Since the records of the ALBEX
and TROL landers obtained during the simultaneous deployment at the deep mound site (~530 m) did
not show significant differences, only the data of the ALBEX lander are here presented (Table 1). These
data are compared with ALBEX lander data obtained during its deployment in the shallow part of the
mound zone at ~340 m (Table 1). Deployment times varied from 2.5 to 8 days (Table 1).
Both, the ALBEX and TROL lander consist of an aluminum tripod for this experiment equipped with 13
glass benthos floats, two IXSEA acoustic releasers and a single 260 kg ballast weight. Oceanographic data
were obtained by different sensors: an ARO-USB oxygen sensor (JFE-Advantech™) which also recorded
temperature, a combined OBS-fluorometer (Wetlabs™) and an Aquadopp (Nortek™) profiling current
meter. The ALBEX lander was furthermore equipped with a Technicap PPS4/3 sediment trap with 12
bottles (allowing daily samples) and a McLane particle pump (24 filter units for each 7.5 L of seawater,
two hour interval) to sample particulate organic matter in the near-bottom water (40 cm above
bottom).
Additionally, a GEOMAR Satellite Lander Module (SLM) was deployed off-mound close to the NIOZ
landers (Fig. 1, Table1). The SLM was equipped with a 600 kHz ADCP Workhorse Sentinel 600 from RDI, a
CTD (SBE SBE16V2™), a combined fluorescence and turbidity sensor (WET Labs ECO-AFL/FL), a dissolved
oxygen sensor (SBE™) and a pH sensor (SBE™) (Hebbeln et al., 2017). From the SLM only pH
measurements are used here, complementing the data from the NIOZ landers.



### 2.2.2 CTD transects

Vertical profiles of principal hydrographic parameters in the watercolumn, viz. temperature, conductivity, oxygen and turbidity, were obtained using a Seabird CTD/Rosette system (Seabird SBE 9 plus). The additional sensors on the CTD were a dissolved oxygen sensor (SBE 43 membrane-type DO Sensor) and a combined fluorescence and turbidity sensor (WET Labs ECO-AFL/FL). The CTD was combined with a rosette water sampler consisting of 24 Niskin® water sampling bottles (10 L) that were electronically triggered to close at given depths during the up-cast. CTD casts were carried out along two downslope CTD transects (Fig. 1). Off Angola, the downslope transect covered a distance of 20 km reaching down to a depth of 800 m, whereas the main transect in Namibia covered a distance of 60 km and maximum depth of 400 m. In order to measure deeper water masses, one deep CTD cast was conducted at a distance of about 130 km from the shallowest CTD, going down to about 1000 m depth (Figs. 1 and 3).

### 2.2.3 Hydrographic data processing

The CTD dataset was processed using the processing software Seabird data SBE 11plus V 5.2 and data were visualized using the program Ocean Data View (Schlitzer 2011; Version 4.7.8). Turbidity data were only collected on the Angolan slope.

Hydrographic data recorded by the CTD and landers were analyzed and plotted using the program R (Team, 2017). Data from the different instruments (temperature, turbidity, current speed, oxygen concentration, fluorescence) were averaged over a period of 1.5 h to remove shorter tem trends and occasional spikes. Correlations between variables were assessed by Spearman's rank correlation tests.

### 2.2.4 Suspended particulate matter

With each upcast of the CTD/Rosette, water samples were taken as close to the seabed as possible, at mid-water depth, and in the chlorophyll-maximum. From each depth, two 5 L water samples were subsequently filtered over pre-combusted (450 °C) and pre-weighted GF/F filters (47 mm, Whatman™). Filters were stored at -20 °C until further analysis at the NIOZ.

Near-bottom suspended particulate organic matter (SPOM) was additionally sampled by means of a phytoplankton sampler (McLane PPS) mounted on the ALBEX lander. The PPS was fitted with 24 GF/F filters (47 mm Whatman™ GF/F filters pre-combusted at 450 °C). A maximum of 7.5 L was pumped over each filter during a 2h period yielding a time series of near bottom SPOM supply and its variability over a period of 48 hours.



*C/N analysis and isotope measurements*


Filters from the in situ pump and sampled from the CTD/Rosette were freeze-dried before further
analysis. Half of each filter was used for phytopigment analysis and a ¼ section of each filter was used
for analyzing organic carbon, nitrogen, and their stable isotope ratios. The filters, used for carbon
analysis, were decarbonized by vapor of concentrated hydrochloric acid (2 M HCl supra) prior to
analyses. Filters were transferred into pressed tin capsules (12x5 mm, Elemental Microanalysis) and
$\delta^{15}$N, $\delta^{13}$C and total weight percent of organic carbon and nitrogen were analyzed by a Delta V
Advantage isotope ratio MS coupled on line to an Elemental Analyzer (Flash 2000 EA-IRMS) by a Conflo
IV (Thermo Fisher Scientific Inc.). The used reference gas was purified atmospheric $N_2$. As a standard for
$\delta^{13}$C benzoic acid and acetanilide was used, for $\delta^{15}$N acetanilide, urea and casein was used. For $\delta^{13}$C
analysis a high signal method was exercised including a 70% dilution. Values are reported relative to v-
pdb and the atmosphere respectively. Precision and accuracy based on replicate analyses and
comparing international standards for $\delta^{13}$C and $\delta^{15}$N was ± 0.15 ‰. The C/N ratio is based on the weight
ratios between TOC and N.

*Phytopigments*


Phytopigments were measured by reverse-phase high-performance liquid chromatography (RP-HPLC)
with a gradient based on the method published by (Kraay et al., 1992). For each sample half of a GF/F
filter was used and freeze-dried before extraction. Pigments were extracted using 95% methanol and
sonification. All steps were performed in a dark and cooled environment. Pigments were identified by
means of their absorption spectrum, fluorescence and the elution time. Identification and quantification
took place as described by Tahey et al. (1994). The absorbance peak areas of chlorophyll-α were
converted into concentrations using conversion factors determined with a certified standard. The
∑Phaeopigment/ Chlorophyll-α ratio gives an indication about the degradation status of the organic
material, since phaeopigments form as a result of bacterial or autolytic cell lysis and grazing activity
(Welschmeyer and Lorenzen, 1985).

*2.2.5 Tidal analysis*


The barotropic (due to the sea level and pressure change) and baroclinic (internal „free waves"
propagating along the pycnoclines) tidal signals obtained by the Aquadopp (Nortek™) profiling current
meter were analysed from the bottom pressure and from the horizontal flow components recorded 6 m
above the sea floor, usinf the harmonic analysis toolbox t_tide (Pawlowicz et al., 2002). The data mean
and trends were subtracted from the data before analysis.



## 3. Results

### 3.1 Namibian margin

*3.1.1 Water column properties off Namibia*

The hydrographic data obtained by CTD measurements along a downslope transect from the surface to 1000 m water depth revealed distinct changes in temperature and salinity through the water column. These are ascribed to the different water masses in the study area (Fig. 3a). In the upper 85 m of the water column, temperatures are above 14°C and salinities are > 35.2, which corresponds to South Atlantic Subtropical Surface Water (SASSW). SACW is situated underneath the SASSW and reaches down to about 700 m.  SCAW is defined by a linear relationship between temperature and salinity in the TS-plot (Shannon et al., 1987). The temperature in the layer of SCAW decreases from 14 to 7°C with depth and the salinity from 35.4 to 34.5 (Fig. 3a). The deep CTD cast about 130 km from the coastline recorded a water mass with the signature of ESACW, having a lower temperature (-1.3°C) and lower salinity (-0.2 PSU) than SACW (in 200 m depth, not included in CTD transects of Fig. 4). Underneath these two central water masses Antarctic Intermediate Water (AAIW) was found with a temperature <7°C.

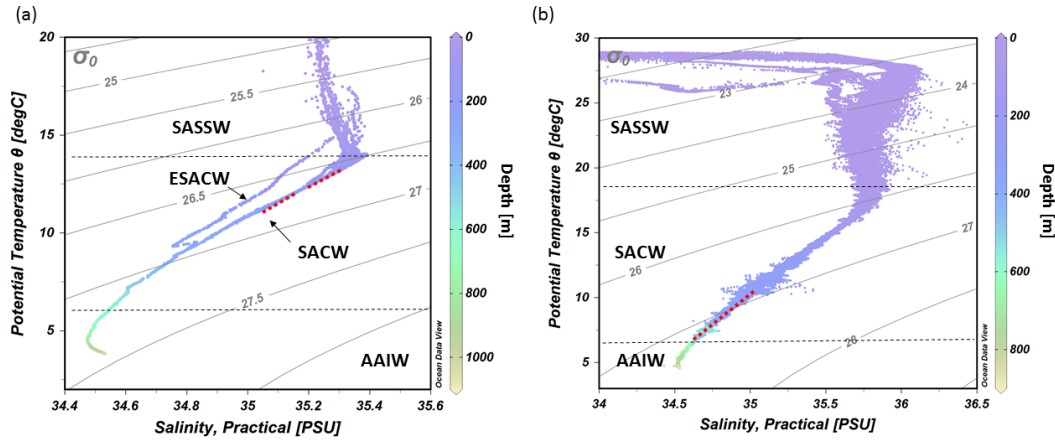

**Figure 3** TS-diagrams showing the different water masses being present at the (a) Namibian and (b) Angolan margins: South Atlantic Subtropical Surface Water (SASSW), South Atlantic Central Water (SACW) and Eastern South Atlantic Central water (ESACW), Antarctic Intermediate Water (AAIW) (data plotted using Ocean Data View v.4.7.8; http://odv.awi.de; Schlitzer, 2011). Red dotted line indicates the depth range of cold-water coral mound occurrence.

The CTD transect showed decreasing $DO_{conc}$ from the surface (6 ml l⁻¹) towards a minimum in 150-200 m depth (0 ml l⁻¹). Lowest values for $DO_{conc}$ were found on the continental margin between 100-335 m water depth. The $DO_{conc}$ in this pronounced OMZ ranged from <1 ml l⁻¹ down to 0 ml l⁻¹ (≙ 9-0 %



saturation, respectively). The zone of low $DO_{conc}$ (<1 ml l$^{-1}$) was stretching horizontally over the complete

transect towards at least 100 km offshore (Fig. 4c). The upper boundary of the OMZ was relatively sharp

compared to its lower limits and corresponds with the border between SASSW at the surface and SACW

below.

Within the OMZ, a small increase in fluorescence (0.2 mg m$^{-3}$) was recorded, whereas fluorescence was

otherwise not traceable below the surface layer (Fig. 4d). Within the surface layer highest surface

fluorescence (>2 mg m$^{-3}$) was found ~40 km offshore. Above the center of the OMZ fluorescence

reached only up to 0.4 mg m$^{-3}$.

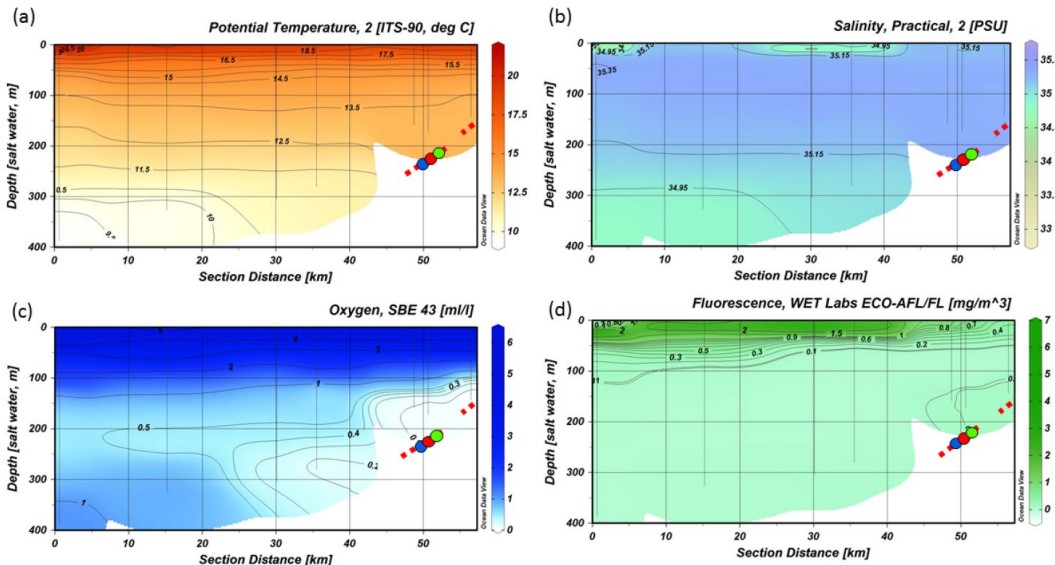

**Figure 4** CTD transect across the Namibian margin. Shown are data for: (a) potential temperature (°C),( b) salinity (PSU), (c)

dissolved oxygen concentrations (ml l$^{-1}$), note the pronounced oxygen minimum zone (OMZ) between 100-335 m water depth,

and d) fluorescence (mg m$^{-3}$) (data plotted using Ocean Data View v.4.7.8; http://odv.awi.de; Schlitzer, 2011). The occurrence of

fossil CWC mounds is indicated by a red dashed line, colored dots indicate bottom lander deployments.

*3.1.2 Lander time-series of physical data*

Bottom temperature ranged from 11.8 to 13.2°C during the deployment of the ALBEX lander (Table 1)

showing oscillating fluctuations with a maximum semidiurnal (Δt ~ 6h) change of ~1°C (on 9.1.2018). The

$DO_{conc}$ fluctuated between 0-0.15 ml l$^{-1}$ and was negatively correlated with temperature (r=-0.39,

p<0.01). Fluorescence ranged from 42 to 45 NTU during the deployment and was positively correlated

with temperature (r=0.38, p<0.01). Hence, both temperature and fluorescence were negatively





correlated with oxygen concentration (r=-0.39, p<0.01) and also with turbidity (optical backscatter, r=-
0.35, p<0.01). Turbidity was relatively low until it increased especially during the second half of the
deployment, when wind speeds increased and also the current direction changed. The maximum current
speeds measured during the deployment period were 0.21 m s$^{-1}$, with average current speeds of 0.09 m
s$^{-1}$ (Table 2). The tidal cycle explains >80 % of the pressure fluctuations (Table 2), with a semidiurnal
signal, M2, generating an amplitude of >0.35 dbar and thus being the most important constituent.
Before the 6$^{th}$ of January the current direction oscillated between SW and SE where after it changed into
a dominating northern current direction. The current speed remained rather constant during the
deployment period (Fig. 5). The wind speed, on the other hand, increased from 10 m s$^{-1}$ to a maximum
of 17 m s$^{-1}$ on the sixth of January and remained high for the next six days. Wind direction changed from
anticlockwise cyclonic rotation towards alongshore winds. The water current direction returned to SW-
SE after the period of strong wind (not shown). During the strong wind period, colder water (correlation
between wind speed and water temperature, r=-0.55, p<0.01), with a higher turbidity (correlation of
wind speed and turbidity, r=0.42, p<0.01) and higher DO$_{conc}$ was present. The SLM lander recorded an
average pH of 8.01.
The observed fluctuations in bottom water temperature at the deployment site imply a vertical tidal
movement of around 70 m. This was calculated by comparing the temperature change recorded by the
lander to the respective temperature-depth gradient based on water column measurements (CTD site
GeoB20553, 12.58 °C at 245 m, 12.93°C at 179 m). Due to these vertical tidal movements, the oxygen
depleted water from the core of the OMZ is regularly being replaced with somewhat colder and slightly
more oxygenated water (up to 0.2 ml l$^{-1}$).






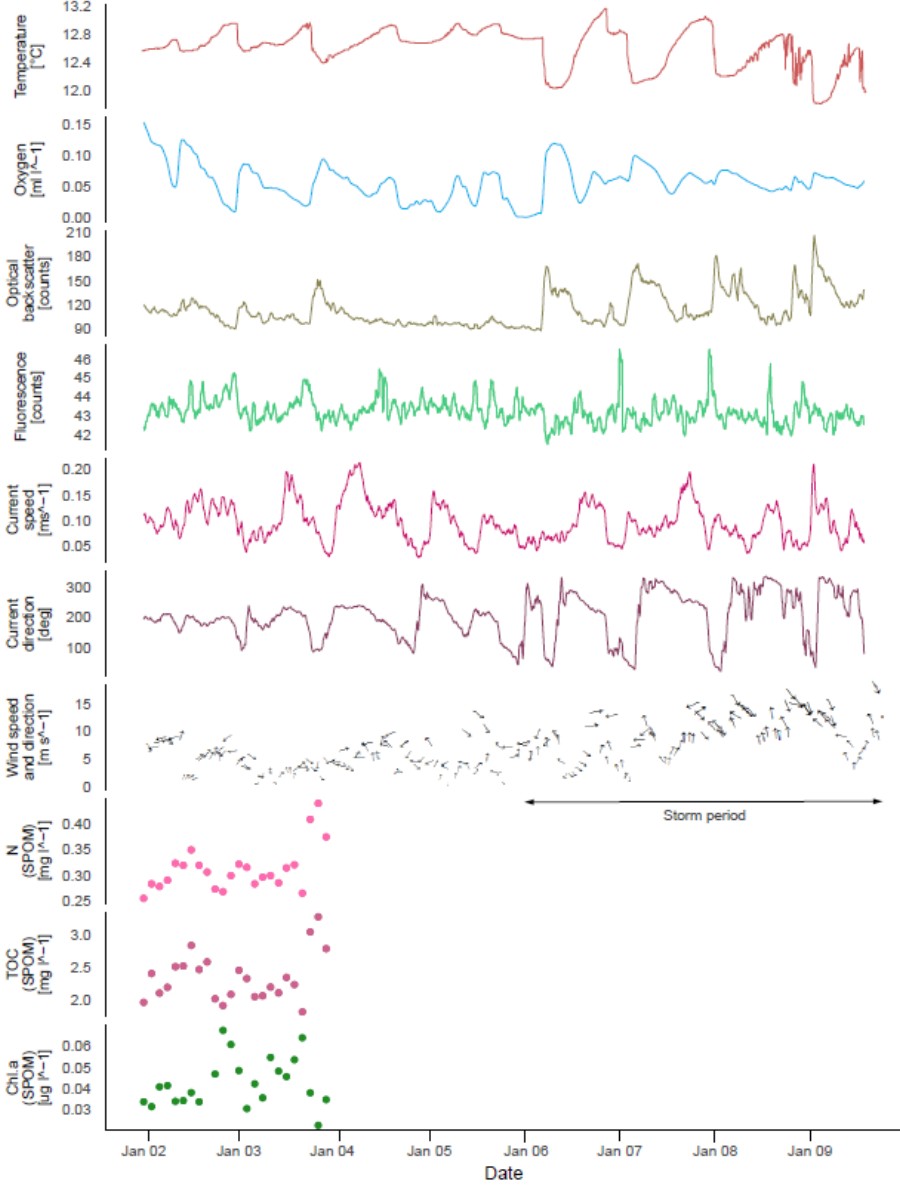

**Figure 5** Data recorded by the ALBEX lander (210 m) at the Namibian margin in January 2016. Shown are data for temperature
(°C; red), dissolved oxygen concentrations (ml l$^{-1}$; blue), optical backscatter (turbidity; moss green), fluorescence (counts per
second green), current speed (m s$^{-1}$; pink), current direction (degree: 0-360°; dark red) as well as nitrogen (mg l$^{-1}$; pink dots),
carbon (mg l$^{-1}$; purple dots), and chlorophyll-α concentration (µg l$^{-1,}$ green dots) of SPOM collected during the first 48h by the
McLane pump. These data are supplemented by wind speed and direction (small black arrows) recorded concurrently to the
lander deployment by ship bound devices. Note that current directions changed from a generally south-poleward to an
equatorward direction when wind speed exceeded 10 m s$^{-1}$ (stormy period indicated by black arrow).



*3.1.3 Food supply*
The nitrogen (N) concentration of the SPOM measured on the filters of the ALBEX lander McLane pump
fluctuated between 0.25 and 0.45 mg l$^{-1}$ (Fig 5). The highest N concentration corresponded with a peak
in turbidity (r=0.42, p<0.01). The $\delta^{15}$N values of the lander time series fluctuated between 5.1 and 6.9
with an average value of 5.7 ‰. Total Organic Carbon (TOC) showed a similar pattern as nitrogen, with
relative concentrations ranging between 1.8-3.5 mg l$^{-1}$. The $\delta^{13}$C value of the TOC increased during the
surveyed time period from -22.39 to -21.24‰ with an average of -21.7 ‰ (Fig. 6a). The C/N ratio ranged
from 8.5 to 6.8 and was on average 7.4 (Fig. 6b). During periods of low temperature and more turbid
conditions TOC and N as well as the $\delta^{13}$C values of the SPOM were higher.
Chlorophyll-α concentrations in the SPOM collected with the lander in situ pump were on average 0.042
µg l$^{-1}$ and correlated with the record of the fluorescence sensor on the Seabird CTD (r=0.43, p=0.04). A
six times higher amount of chlorophyll-α degradation products were found during the lander
deployment (0.248 µg l$^{-1}$) compared to the amount of chlorophyll-α, giving a ∑Phaeopigment/
Chlorophyll-α ratio of 6.5 (not shown). Additionally, carotenoids (0.08-0.12 µg l$^{-1}$) and fucoxanthin (0.22
µg l$^{-1}$) were found as major components of the pigment fraction, which are common in diatoms.
Zeaxanthin, indicating the presence of prokaryotic cyanobacteria, was only observed in small quantities
in the SPM (0.066 µg l$^{-1}$).

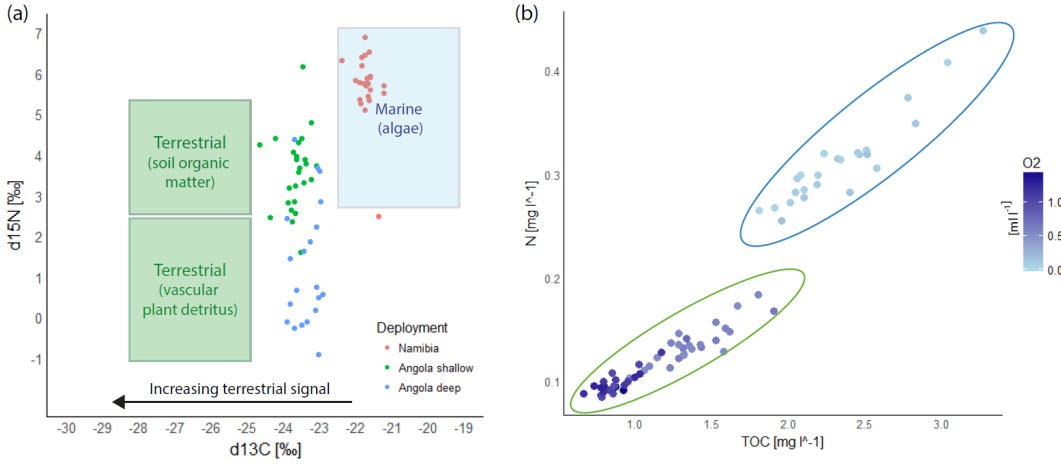


**Figure 6** Composite records of SPOM collected by the McLane pump of the ALBEX lander at the Namibian and Angolan margins
during all three deployments. (a) δ15N and δ13C isotopic values at the Namibian (red dots) and Angolan (blue and green dots)
margins. Indicated by the square boxes are common isotopic values of terrestrial and marine organic matter (Boutton 1991,
Holmes et al. 1997, Sigman et al. 2009). The relative contribution of terrestrial material (green boxes) is increasing with a more




negative δ13C value. (b) Total organic carbon (TOC) and nitrogen (N) concentration of the SPOM. Values of the Namibian
margin are marked by a blue circle (C/N ratio = 7.8), values of the Angolan margin are marked by a green circle (C/N ratio = 9.6).
Dissolved oxygen concentrations are included to show the higher nutrient concentrations in less oxygenated water.

## 3.2 Angolan margin

### 3.2.1 Water column properties

The hydrographic data obtained by CTD measurements along a downslope transect from the surface to
800 m water depth revealed distinct changes in temperature and salinity throughout the water column,
related to four different water masses. At the surface a distinct shallow layer (>20 m) with a distinctly
lower salinity (27.3-35.5) and higher temperature (29.5-27 °C, Fig. 3b) was observed. Below the surface
layer, SASSW was recognized down to a depth of 70 m, characterized by a higher salinity (35.8). SACW
was observed between 70-600 m, featuring the expected linear relationship between temperature and
salinity. Temperature and salinity decreased from 17.5°C/35.8 to 7°C/34.6. At 700 m depth AAIW was
recorded, characterized by a low salinity (<34.4) and temperature (<7°C, Fig. 3b).
The CTD transect shows a sharp decrease in the $DO_{conc}$ underneath the SASSW from 5 to <2 ml l$^{-1}$ (Fig. 7).
$DO_{conc}$ was further decreasing until a minimum of 0.6 ml l$^{-1}$ at 350 m and subsequently increasing to >3
ml l$^{-1}$ at 800 m depth. Lowest $DO_{conc}$ were not found at the slope but 70 km offshore in the center of the
zone of reduced $DO_{conc}$ between 200-450 m water depth (<1 ml l$^{-1}$). Compared to the Namibian margin
(see Fig. 4), the hypoxic layer was hence situated further offshore, slightly deeper and overall $DO_{conc}$
were higher (compare Fig. 4c). Also, the boundaries of the hypoxic zone were not as sharp. Salinity
underneath the surface layer decreased linearly from 35.75 to 34.5 in 800 m and did not show any
specific features likewise as the temperature which decreased from 16 to 5°C. Fluorescence near the sea
surface was generally low (around 0.2 with small maxima of 0.78 mg m$^{-3}$) and not detectable deeper
than 150 m depth. The OBS signal showed a distinct zone of enhanced turbidity on the continental
margin between 200-350 m water depth.



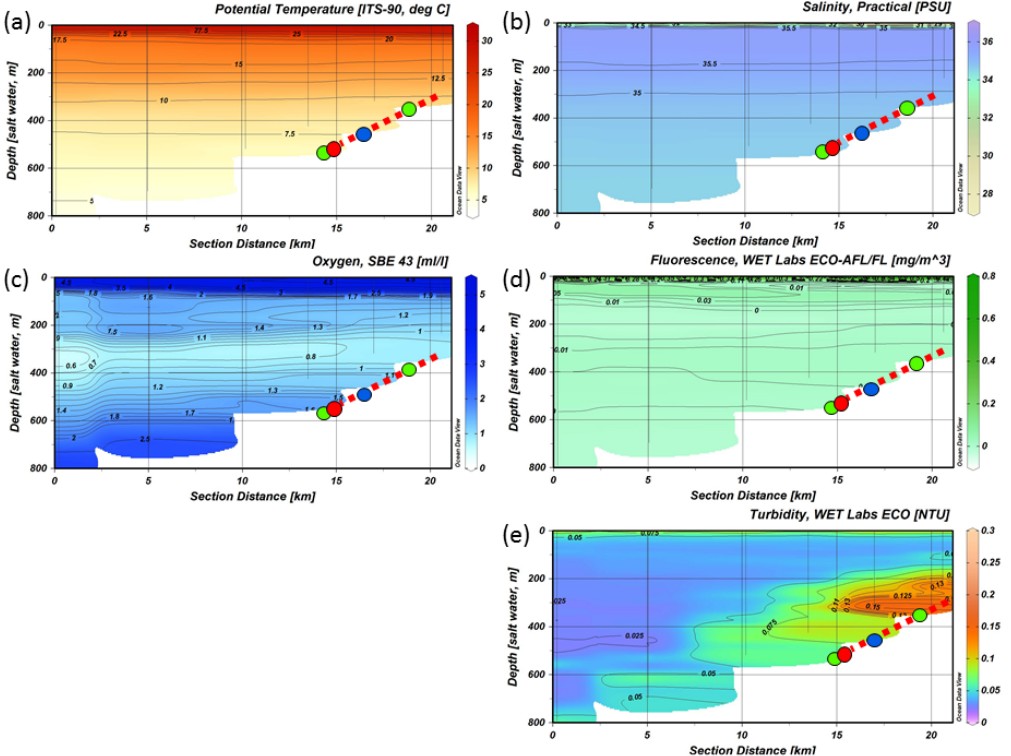

**Figure 7** CTD transect across the Angolan margin. Shown are data for (a) potential temperature (°C), (b) salinity (PSU), (c) dissolved oxygen concentration (ml l⁻¹), (d) fluorescence (mg m⁻³), (e) turbidity (NTU) (data plotted using Ocean Data View v.4.7.8; http://odv.awi.de; Schlitzer, 2011). The depth occurrence of CWC mounds is marked by a red, dashed line, the lander deployments are indicated by colored dots.

### 3.2.2 Lander time series physical data

Mean bottom water temperatures varied from the deepest to the shallowest site between 6.73-10.06 °C (Fig. 8). The maximum semidiurnal ($\Delta t \sim 6h$) temperature change varied between 0.82 to 1.60 °C at the deepest and shallowest site respectively. At the shallow site, the maximum short-term temperature change was 2.4 °C (Fig. 8). $DO_{conc}$ at the deep site were a factor of two higher than those at the shallow site, i.e. 0.9-1.5 $vs$. 0.5-0.8 ml l⁻¹respectively ($\triangleq$ range between 4-14% saturation of both sites), where also the range of diurnal fluctuations was much smaller compared to the shallow site. $DO_{conc}$ were negatively correlated with temperature at the deep site (r=-0.99, p<0.01) while positively correlated at the shallow site (r=0.91, p<0.01). Fluorescence was overall low during both deployments and showed only small fluctuations whereas it was slightly higher at the shallow site (between 38.5 and 41.5 NTU at both sites). Current speeds were relatively high (between 0–0.3 m s⁻¹, average 0.1 m s⁻¹) and positively





correlated with temperature at the shallow site (r=0.31, p<0.01) and negatively correlated at the deep
site (r=-0.22, p<0.01). Analysis of the tidal cycle showed, that it explains 29.8-54.9% of the horizontal
current fluctuations. The M2 (principal lunar semi-diurnal) amplitude was 0.06-0.09 m s$^{-1}$ and was the
most important signal (Table 2). The OBS measurements showed a decreasing turbidity during the
deployment at the shallow station. This station was located directly below the turbidity maximum
between 200-350 m depth as observed in the CTD transect (Fig.7). In contrast, a relative constant and
low turbidity was observed for the deep deployment. Turbidity during both deployments was positively
correlated to $DO_{conc}$ (r=0.47, p<0.01, shallow and r=0.50, p<0.01, deep). The SLM lander was deployed at
a depth of 440 m and recorded an average pH of 8.12.
The short-term temperature fluctuations imply a vertical tidal movement of around 130 m, based on
comparing the temperature range measured by the lander and the temperature versus depth gradient
based on CTD measurements (12.9-9.1 °C measured by lander ≙ 218-349 m depth in CTD above lander
at station GeoB20966).




**Figure 8** Lander data (ALBEX) recorded during the shallow (~340 m water depth) and deep deployments (~530 m water depth)
off Angola (January 2016). Shown are temperature (°C; red), dissolved oxygen concentration (ml l⁻¹; blue), fluorescence (counts
per second; green), optical backscatter (turbidity; yellow), current speed (m s⁻¹; pink) and current direction (degree: 0-360°;
purple) as well as nitrogen (mg l⁻¹; pink dots), carbon (mg l⁻¹; purple dots), and chlorophyll-α concentration (µg l⁻¹, green dots) of
SPOM collected during the both deployments by the McLane pump.



*3.2.3 Food supply*
In general TOC and N concentrations of SPOM measured on the filters from the McLane pump were
lower at the deep site. Nitrogen concentrations varied around 0.14 mg l$^{-1}$ at 342 m and around 0.1 mg l$^{-1}$
at 532 m depth (Fig. 6b). The $\delta^{15}$N values of the lander time series of the shallow deployment ranged
from 1.6 to 6.2 ‰ (3.7 ‰ average) and were even lower deeper in the water column, viz. range 0.3-3.7
‰ with an average of 1.4 ‰. The TOC concentrations were on average 1.43 mg l$^{-1}$ at 342 m and 0.9 mg l$^{-}$
$^{1}$ at 532 m, with corresponding $\delta^{13}$C values ranging between -23.0 and -24.2 (average of -23.6 ‰) at the
shallow and between -22.9 and -23.9 (average -23.4 ‰) at the deep site. The C/N ratio was relatively
stable and on average 10.2 and 9 for the samples taken at the shallower and deeper site, respectively.
The chlorophyll-α concentrations of the SPOM collected by the McLane pump varied between 0.1 and
0.02 µg l$^{-1}$, with an average phaeopigments/chlorophyll-α ratio of 2.6 and 0.5 on the shallow and deep
site, respectively. Phytopigments recorded by the shallow deployment included 0.3 µg l$^{-1}$ of fucoxanthin,
while at the deep site only a concentration of 0.1 µg l$^{-1}$ was found. No zeaxanthin was recorded in the
pigment fraction.
# 4. Discussion
Even though the ecological-niche factor analysis of Davies et al. (2008) and Davies and Guinotte (2011)
predict *L. pertusa* to be absent on the oxygen-limited southwestern African margin, two CWC mound
ecosystems were observed along the Namibian and Angolan margins. The coral mounds on the Namibian
shelf host no living CWCs, instead dead coral framework covering the mounds was overgrown with fauna
dominated by bryozoans and sponges. Along the slope of the Angolan margin an extended coral mound
area with thriving CWC communities was encountered. Differences between the areas indicate different
environmental conditions influencing faunal assemblages present in both areas. The potential impact of
the key environmental factors will be discussed below.
*4.1 Short-term vs long-term variations in environmental properties*
On the Namibian margin, seasonality has a major impact on local-mid-depth oxygen concentration due
to the periodically varying influence of the Angola current and its associated low DO$_{conc}$. The lowest
DO$_{conc}$ are expected from February to May when SACW is the dominating water mass on the Namibian
margin and the contribution of ESAC water is smaller. Due to this seasonal pattern, the DO$_{conc}$ measured
in this study (January; Figs. 4) most likely do not represent minimum concentrations, which are expected
to occur in the following months (February to May; Mohrholz et al., 2014). Interestingly, we captured a
flow reversal from a southward to an equatorward current direction during high wind conditions on the



Namibian margin (Fig. 5), leading to an intrusion of ESACW with higher $DO_{conc}$ (+0.007 ml l$^{-1}$ on average)
and lower temperatures (-0.23°C on average, Fig. 5) than the SACW after the 6$^{th}$ of January, leading to
an temporal relaxation of the oxygen stress. This shows that variations in the local flow field have the
capability to change water properties on relatively short time scales, which might provide an analogue
to the water mass variability related to the different seasons. Such relaxations are likely important for
the survival of the abundant invertebrate fauna present on the relict coral mounds under the conditions
generally considered unsuited for them. Other seasonal changes, like riverine outflow do not have
decisive impacts on the margin ecosystem since only relatively small rivers discharge from the Namibian
margin. This is also reflected by the dominant marine isotopic signature of the isotopic ratios of δ$^{15}$N and
δ$^{13}$C of the suspended particulate matter at the mound areas (Fig. 6, cf. Tyrrell and Lucas, 2002).
Flow reversals were not observed during the lander deployments on the Angolan margin, where winds
are reported to be weak throughout the year providing more stable conditions (Shannon, 2001). Instead
river outflow seems to exert a strong influence on the oxygen concentration on the Angolan margin. The
run-off of the Cuanza and Congo river reach their seasonal maximum in December and January (Kopte et
al., 2017), intensifying upper water column stratification and transporting terrestrial organic matter to
the margin. This stratification is restricting vertical mixing and thereby limits ventilation of the oxygen
depleted subsurface water masses. The input of terrestrial organic matter is reflected by the isotopic
signals of the SPOM, i.e. a δ$^{15}$N values between 1.4 to 3.7‰. This range resembles a more terrestrial
signal (-1 to 3‰; Montoya, 2007) and is well below the average isotopic ratio of the marine waters of
5.5‰ (Meisel et al., 2011). Also δ$^{13}$C values of -23.5‰ are in line with the δ$^{13}$C values of terrestrial
matter which is on average -27 ‰ in this area (Boutton, 1991;Mariotti et al., 1991). The C/N ratio of
SPOM is higher compared to material from the Namibian margin, confirming admixing of terrestrial
matter (Perdue and Koprivnjak, 2007). The combined effects of decreased vertical mixing and additional
input of organic matter potentially result in the lowest $DO_{conc}$ of the year during the investigated time
period (January), since the highest river outflow and therefore strongest stratification is expected during
this period.

### 4.2 Main stressors – Oxygen and temperature

Environmental conditions marked by severe hypoxia and temporal anoxia (<0.17 ml l$^{-1}$) likely explain the
present-day absence of living CWCs along the Namibian margin. During the measurement period the
$DO_{conc}$ off Namibia were considerably lower than the thus far recorded minimum concentrations near
living CWCs (1-1.3 ml l$^{-1}$), which were found off Mauritania where only isolated living CWCs are found



(Ramos et al., 2017). Age dating of  the Namibian fossil coral framework shows that CWCs disappeared
about 5 ka BP, which coincides with an intensification in upwelling and therefore most likely a decline of
$DO_{conc}$ (Tamborrino et al., sumbitted). This is supporting the assumption that the $DO_{conc}$ is responsible for
the demise of CWCs on the Namibian margin. Although no living corals were observed on the Namibian
coral mounds, we observed a dense living community dominated by sponges and bryozoans (Hebbeln et
al., 2017). Several sponge species have been reported to survive at extremely low $DO_{conc}$ within OMZs.
For instance, along the lower boundary of the Peruvian OMZ sponges were found at $DO_{conc}$ as low as
0.06-0.18 ml $l^{-1}$ (Mosch et al., 2012). Mills et al. (2018) recently found a sponge (*Tethya wilhelma*) to be
physiologically almost insensitive to oxygen stress and to respire aerobically under low $DO_{conc}$ (0.02 ml $l^{-1}$
). Sponges can potentially stop their metabolic activity during unfavorable conditions and re-start their
metabolism when some oxygen becomes available, for instance during diurnal irrigation of water with
somewhat higher $DO_{conc}$. The existence of a living sponge community off Namibia might hence be
explained by the diurnal baroclinic tides occasionally flushing the sponges with more oxic water enabling
them to metabolize, when food availability is also highest (pulse of suspended particulate matter with a
higher amount of TOC and N during oxygenated conditions, Figs. 5, 8). Increased biomass and
abundances in these temporary hypoxic-anoxic transition zones were already observed for macro- and
mega-fauna in other OMZs and is referred to as the "edge effect" (Mullins et al., 1985;Levin et al.,
1991;Sanders, 1969). It is very likely that this mechanism plays a role for the benthic communities on the
Angolan margin.
Along the Angolan margin low oxygen concentrations apparently do not restrict the proliferation of
thriving CWC reefs even though $DO_{conc}$ are considered hypoxic (0.5-1.5 ml $l^{-1}$). The $DO_{conc}$ measured off
Angola are well below the lower $DO_{conc}$ limits for *L. pertusa* based either on habitat suitability modelling
(Davies et al., 2008) or on laboratory experiments and earlier field observations (Schroeder, 2002;Brooke
and Ross, 2014). The $DO_{conc}$ encountered at the shallow mound sites (<0.8 ml $l^{-1}$) are even below the so far
lowest limits known for single CWC colonies from the Mauritanian margin (Ramos et al., 2017b). Since in
the present study, measured $DO_{conc}$ were even lower than the earlier established lower limits3 this could
suggest at a much higher tolerance of *L. pertusa* to low oxygen levels at least in a limited time-period as
low as 0.5 ml $l^{-1}$ (4% $O_2$ saturation), which was measured by the ALBEX lander as well as the CTD during
the cruise in 2016. Even though concentrations at the Angolan margin showed to be relatively stable, it
should be emphasized that the observation period was limited and long-term (year-long) observations
remain necessary to confidently extend the lower limit of oxygen deficiency tolerance by *L. pertusa*.



In addition to the oxygen stress, heat stress is expected to put additional pressure on CWCs. Temperatures
at the CWC mounds off Angola ranged from 6.4 to 12.6 °C, which are close to their reported maximum
temperatures (~12-14.9 °C; Davies and Guinotte 2011) and are hence expected to impair the ability of
CWCs to form mounds (see Wienberg and Titschack 2017). In most aquatic invertebrates respiration rates
roughly double with every 10 °C increase ($Q_{10}$ temperature coefficient = 2-3, e.g. Coma 2002), which at
the same time doubles energy demand. Dodds et al. (2007) found a doubling of the respiration rate of *L.*
*pertusa* with an increase at ambient temperature of only 2 °C (viz. $Q_{10}$=7-8). This would limit the survival
of *L. pertusa* at high temperatures to areas where the increased demand in energy (due to increased
respiration) can be compensated by high food availability. Higher respiration rates also imply that enough
oxygen needs to be available for the increased respiration.
Survival of *L. pertusa* under hypoxic conditions along the shallow Angolan CWC areas is probably positively
influenced by the fact that periods of highest temperatures coincide with highest $DO_{conc}$ during the tidal
cycle, which stands in contrast to mounds on the Namibian margin or the deeper Angolan mound area.
Probably the increase of one stressor is compensated by a reduction of another stressor in the shallow
Angolan mound areas. On the Namibian margin or the deeper Angolan mound sites we found the opposite
pattern, with highest temperatures during lowest $DO_{conc}$. The occurrence of *L. pertusa* at the the deeper
Angolan mound sites is possibly related to the fact that $DO_{conc}$ are anyway higher and temperatures more
within a suitable range compared to the shallow sites (0.9-1.5 ml l$^{-1}$, 6.4-8 °C, Fig. 8). Additionally it was
shown by ex situ experiments that *L. pertusa* is able to survive periods of hypoxic conditions similar to
those found along the Angolan margin for several days, which could be crucial in periods of most adverse
conditions during one tidal period or also slightly longer time periods (Dodds et al., 2007). However,
oxygen stress leads to a loss of energy which and associated increased energy demand like in other aquatic
invertebrates (Sokolova et al., 2012).

### 4.3 Food supply

As mentioned above, environmental stress like high temperature or low $DO_{conc}$ result in a loss of energy
(Odum, 1971;Sokolova et al., 2012), which needs to be balanced by an increased energy (food) availability.
Food availability therefore plays a significant role for faunal abundance under hypoxia or unfavorable
temperatures (Diaz and Rosenberg, 1995). Above, we argued that survival of sponges and bryozoans on
the relict mounds off Namibia and of CWCs and their associated fauna at the Angolan margin*,* may be
partly due to a high input of high-quality organic matter, compensating the oxygen and thermal stresses.
The importance of the food availability for CWCs was already suggested by Eisele et al. (2011), who



mechanistically linked CWC mound growth periods with enhanced surface water productivity and hence
organic matter supply. Here we found evidence for high quality and quantity of SPOM in the vicinity of
the coral mounds on the Namibian as well as on the Angolan margin. Indicators for the high quality of
food in the SPOM at both sites were high TOC and N concentrations (Figs. 6, 10) in combination with a
low C/N ratio (Fig. 7), a low isotopic signature of $\delta^{15}N$ and only slightly degraded pigments
(∑phaeopigment/ chlorophyll-α ratio of on average 6.5 and 3.2 off Namibia and Angola, respectively).
The Namibian margin is known for its upwelling cells, where phytoplankton growth is fueled by nutrients
from deeper water layers producing high amounts of phytodetritus (Chapman and Shannon, 1985), which
subsequently sinks down to the relict mounds on the slope. This high flux and accumulation of fresh SPOM
towards the reefs is evident as a slightly increased fluorescence deeper in the water column around the
mound sites (Fig. 4d). Furthermore, the increased fluorescence and chlorophyll-α concentrations coincide
with low $DO_{conc}$ which both are in line with downward movement of waters with a lower $DO_{conc}$ from the
OMZ above (Fig. 5). The higher turbidity during lower current speeds provides additional evidence that
the material settling from the surface is not transported away with the strong currents (Fig. 5). Mounds
off Namibia occur at relatively shallow depths, hence downward transport of SPOM from the surface
waters is rapid and hence time for decomposition of the sinking particles in the water column is limited.
This fast delivery of SPOM over short depth intervals appears to be linked to both, high primary
productivity (high surface fluorescence, Fig. 4d) and reduced decomposition due to low oxygen
concentrations (Pichevin et al., 2004;Cavan et al., 2017), leads to a generous food supply for several
benthic organisms thus enabling them to survive under hypoxic conditions.
At the Angolan coral mounds, SPOM appeared to have a signature corresponding to higher quality organic
matter compared to off Namibia. The phytopigments were less degraded (i.e. higher chlorophyll-α
concentration, lower ∑phaeopigment/ chlorophyll-α ratio) and the $\delta^{15}N$, TOC and N concentration of the
SPOM was lower than off Namibia. However, here lower $\delta^{15}N$ and higher ∑phaeopigment/ chlorophyll-α
ratio are likely connected to terrestrial OM input, which constitutes a less suitable food source for CWCs.
On the other hand the riverine input delivers nutrients, which can support the growth of phytoplankton,
indirectly influencing food supply (Kiriakoulakis et al., 2007;Mienis et al., 2012). Moreover the food quality
at the shallow Angolan reefs was not coupled to periods of other environmental stressors and variations
were relatively small during this study. At the Angolan margin we see a rather constant delivery of SPOM.
The slightly higher turbidity during periods of highest $DO_{conc}$, (Fig. 8) suggest that the SPOM on the Angolan
margin originates from the bottom nepheloid layer on the margin directly above the CWC mounds (Fig.



7e), which may represents a constant reservoir of fresh SPOM. This again indicates that CWCs can benefit
from a constant source of high quality SPOM on the margin somewhat above the coral mound areas and
are not exclusively depending on SPOM settling from the surface (like at the Namibian margin), since the
strong stratification inhibits mixing of the different water masses. This is also supported by the fact, that's
increased fluorescence off Angola was not as strongly correlated with the downward currents as off
Namibia.
*4.4 Tidal currents*
The semidiurnal tidal currents observed probably play a major role in the survival of benthic fauna on the
southwestern African margin. On the Namibian margin internal waves deliver oxygen from the surface
and deeper waters semi-diurnally about 70 m towards to the inside the OMZ and thereby enable benthic
fauna on the fossil coral framework to survive in hypoxic conditions (Fig. 9a). At the same time these
currents are likely responsible for the delivery of fresh SPOM from the surface productive zone to the
communities on the margin.
On the Angolan margin internal tides produce slightly faster currents and vertical excursions of up to 130
m which are twice as high as those on the Namibian margin. Similar to the Namibian margin these tidal
excursions deliver oxygen from shallower and deeper waters to the mound zone (Fig. 9b). Internal tides
are also responsible for the formation of a bottom nepheloid layer in 200-350 m depth (Fig. 7e). This layer
is formed by bottom erosion due to the intensification of near-bottom water movements, which is
indicated by maxima of the buoyancy frequency $N^2$ in 225 and 300 m depth (GeoB20977-1, not shown).
Tidal waves will be amplified due to a critical match between the characteristic slope of the internal M2
tide and the bottom slope of the Angolan margin, as is known from other continental slope regions
(Dickson and McCave, 1986;Mienis et al., 2007). As argued above, this turbid layer is likely important for
the nutrition of the slightly deeper situated CWC mounds, since vertical mixing is hindered by the strong
stratification.



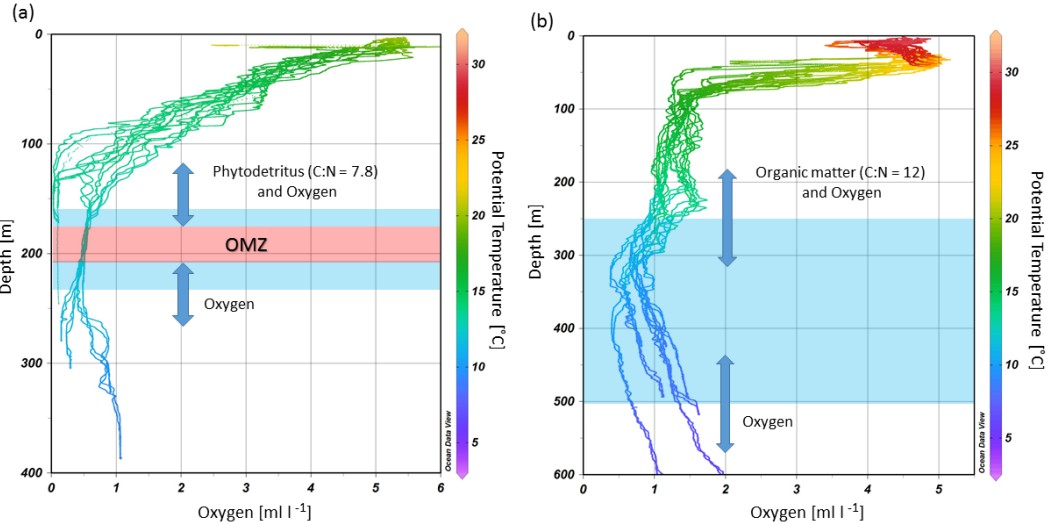


**Figure 9** Depth range of cold-water coral mound occurrences (blue shaded areas) at the (a) Namibian and (b) Angolan margins
in relation to the dissolved oxygen concentrations and potential temperature. Diurnal tides are delivering mainly phytodetritus
(shown in (a) and organic matter from the benthic nepheloid layer (shown in (b) as well as oxygen from above, and from below
to the mound sites (indicated by blue arrows, the length of which indicate the tidal ranges).

### 4.5 Implications

CWC and sponge communities are known to play an important role as a refuge, feeding ground and
nursery for commercial fishes (Miller et al., 2012) and have a crucial role in the marine benthic pelagic
coupling (Cathalot et al., 2015). CWCs and their ecosystem services are threatened by the expected
expansion of OMZs due to anthropogenic activities like rising nutrient loads and climate change
(Breitburg et al., 2018). This study showed that CWCs are able to cope with low oxygen levels as long as
sufficient high quality food is available. Further, reef associated sponge grounds, as encountered on the
Namibian margin could play a crucial role in taking over the function of CWCs in marine carbon cycling
as well as in providing a habitat for associated fauna, when conditions become unsuitable for CWCs.

## 5. Conclusions

Different environmental properties and different relations between these properties explain the
dissimilar present conditions of CWCs on the southwestern African margin including temperature,
$DO_{conc}$, food supply and tidal movements. The $DO_{conc}$ likely defines the state of the CWCs along the
Namibian and the Angolan margin, whereas high temperatures constitute an additional stressor by
increasing the respiration rate and therefore energy demand. On the Namibian margin, where $DO_{conc}$



dropped below 0.01 ml $l^{-1}$, only fossil CWC mounds covered by a community dominated by sponges and
bryozoans were found. This community survives as it receives periodically waters with slightly higher
$DO_{conc}$ (>0.03 ml $l^{-1}$) due to regular tidal oscillations (semi diurnal) and erratic wind events (seasonal). At
the same time, a high quality and quantity of SPOM sinking down from the surface water mass enables
the epifaunal community to survive despite the oxygen stress and sustain its metabolic energy demand
at the Namibian OMZ, while CWCs are not capable to withstand such extreme conditions. In contrast,
thriving CWCs on the Angolan coral mounds were encountered despite the overall hypoxic conditions.
The $DO_{conc}$ were slightly higher than those on the Namibian margin, but nevertheless below the lowest
threshold that was so far reported for *L. pertusa* (Ramos et al., 2017;Davies et al., 2010;Davies et al.,
2008). In combination with temperatures, close to the upper limits for *L. pertusa*, metabolic energy
demand probably reached a maximum. High energy requirements might have been compensated by the
general high availability of fresh resuspended SPOM. Fresh SPOM is accumulated on the Angolan margin
just above the CWC area and is regularly supplied due to mixing by semidiurnal tidal currents, despite
the restricted sinking of SPOM from the surface due to the strong stratification.

## 6. Data availability

Data will be uploaded to Pangea after publication.

## 7. Author contribution

UH analyzed the physical and chemical data, wrote the manuscript and prepared the figures with
contributions of all authors. FM, GD and ML designed the lander research. DH and CW led the cruise and
wrote the initial cruise plan. FM and ML collected the data during the research cruise. WCD was
responsible for water column measurements with the CTD. AF and ML provided habitat characteristics,
including species identification of both CWC areas. KJ performed the tidal analysis and provided
together with SF data of the SML lander. All authors contributed to the data interpretation and
discussion of the manuscript.

## 8. Competing interests

The authors declare that they have no conflict of interest.





## 9. Acknowledgements


We thank the Captain of the RV *Meteor* cruise M122, Rainer Hammacher, his officers and crew, which
contributed to the success of this cruise. We also like to thank the scientific and technical staff for their
assistance during the cruise and the work in the laboratory. Greatly acknowledged are the efforts from
the German Diplomatic Corps in the German Embassies in Windhoek and Luanda and in the Foreign
Office in Berlin. We thank the German Science Foundation (DFG) for providing ship time on RV *Meteor*
and for funding the ROV *Squid* operations to investigate the cold-water coral ecosystems off Angola and
Namibia. This work was further supported through the DFG Research Center/ Cluster of Excellence
"MARUM – The Ocean in the Earth System". UH is funded by the SponGES project, which received
funding from the European Union's Horizon 2020 research and innovation program under grant
agreement No 679849. FM is supported by the Innovational Research Incentives Scheme of the
Netherlands Organisation for Scientific Research (NWO-VIDI grant 016.161.360). GJR is supported by the
Netherlands Earth System Science Centre (NESSC), financially supported by the Ministry of Education,
Culture and Science (OCW). We also thank the Norwegian Research Council (NRC) for funding. KJ is
funded through the FATE project (Fate of cold-water coral reefs – identifying drivers of ecosystem
change).

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

## 11. Figure captions

**Figure 1** (a) Overview map showing the research areas off Angola and Namibia (red squares) and main features of the surface
water circulation (arrows) and frontal zone (dashed line) in the SE Atlantic as well as the two main rivers discharging at the



Angolan margin. Detailed bathymetry maps of the Angolan (upper maps) and Namibian margins (lower maps) showing the
position of (b) CTD transects (note the deep CTD cast down to 1000 m water depth conducted off Namibia) and (c) bottom
lander deployments (red squares shown in b indicate the cutouts displayed in c).
**Figure 2** ROV images (copyright MARUM ROV SQUID, Bremen, Germany) showing the surface coverage of cold-water coral
mounds discovered off Namibia (a, b) and Angola (c, d). Images were recorded and briefly described for their faunal
composition during RV *Meteor* cruise M122 "ANNA" (see Hebbeln et al. 2017). (a) Sylvester mound, 225 m water depth. Dead
coral framework entirely consisting of *Lophelia pertusa*. The framework is intensely colonized by the yellow bryozoan
*Metropriella* sp., zoanthids, actiniarians and sponges. Vagile fauna consists of asteroids and gobiid fishes (*Sufflogobius*
*bibarbatus*) that hide between hollows underneath the coral framework. (b) Sylvester mound, 238 m water depth. Dense coral
rubble (*L. pertusa*) heavily overgrown by *Metropriella* sp. and sponges. Note the decapod crab *Macropipus australis* (center of
the image). (c) Valentine mound, 238 m water depth. Live *Lophelia* colony being grazed by echinoids. Note the sponge
*Aphrocallistes* sp. with its actiniarian symbionts (right side of the image). (d) Buffalo mound, 345 m water depth. Living CWC
reef observed on top of an Angolan coral mound. Many fishes are present around the reef (*Helicolenus dactylopterus*,
*Gephyroberyx darwinii*).
**Figure 3** TS-diagrams showing the different water masses being present at the (a) Namibian and (b) Angolan margins: South
Atlantic Subtropical Surface Water (SASSW), South Atlantic Central Water (SACW) and Eastern South Atlantic Central water
(ESACW), Antarctic Intermediate Water (AAIW) (data plotted using Ocean Data View v.4.7.8; http://odv.awi.de; Schlitzer, 2011).
Red dotted line indicates the depth range of cold-water coral mound occurrence.
**Figure 4** CTD transect across the Namibian margin. Shown are data for: (a) potential temperature (°C),( b) salinity (PSU), (c)
dissolved oxygen concentrations (ml l$^{-1}$), note the pronounced oxygen minimum zone (OMZ) between 100-335 m water depth,
and d) fluorescence (mg m$^{-3}$) (data plotted using Ocean Data View v.4.7.8; http://odv.awi.de; Schlitzer, 2011). The occurrence of
fossil CWC mounds is indicated by a red dashed line, colored dots indicate bottom lander deployments.
**Figure 5** Data recorded by the ALBEX lander (210 m) at the Namibian margin in January 2016. Shown are data for temperature
(°C; red), dissolved oxygen concentrations (ml l$^{-1}$; blue), optical backscatter (turbidity; moss green), fluorescence (counts per
second green), current speed (m s$^{-1}$; pink), current direction (degree: 0-360°; dark red) as well as nitrogen (mg l$^{-1}$; pink dots),
carbon (mg l$^{-1}$; purple dots), and chlorophyll-α concentration (μg l$^{-1}$, green dots) of SPOM collected during the first 48h by the
McLane pump. These data are supplemented by wind speed and direction (small black arrows) recorded concurrently to the
lander deployment by ship bound devices. Note that current directions changed from a generally south-poleward to an
equatorward direction when wind speed exceeded 10 m s$^{-1}$ (stormy period indicated by black arrow).
**Figure 6** Composite records of SPOM collected by the McLane pump of the ALBEX lander at the Namibian and Angolan margins
during all three deployments. (a) δ15N and δ13C isotopic values at the Namibian (red dots) and Angolan (blue and green dots)
margins. Indicated by the square boxes are common isotopic values of terrestrial and marine organic matter (Boutton 1991,
Holmes et al. 1997, Sigman et al. 2009). The relative contribution of terrestrial material (green boxes) is increasing with a more
negative δ13C value. (b) Total organic carbon (TOC) and nitrogen (N) concentration of the SPOM. Values of the Namibian
margin are marked by a blue circle (C/N ratio = 7.8), values of the Angolan margin are marked by a green circle (C/N ratio = 9.6).
Dissolved oxygen concentrations are included to show the higher nutrient concentrations in less oxygenated water.



**Figure 7** CTD transect across the Angolan margin. Shown are data for (a) potential temperature (°C), (b) salinity (PSU), (c)
dissolved oxygen concentration (ml l$^{-1}$), (d) fluorescence (mg m$^{-3}$), (e) turbidity (NTU) (data plotted using Ocean Data View
v.4.7.8; http://odv.awi.de; Schlitzer, 2011). The depth occurrence of CWC mounds is marked by a red, dashed line, the lander
deployments are indicated by colored dots.
**Figure 8** Lander data (ALBEX) recorded during the shallow (~340 m water depth) and deep deployments (~530 m water depth)
off Angola (January 2016). Shown are temperature (°C; red), dissolved oxygen concentration (ml l$^{-1}$; blue), fluorescence (counts
per second; green), optical backscatter (turbidity; yellow), current speed (m s$^{-1}$; pink) and current direction (degree: 0-360°;
purple) as well as nitrogen (mg l$^{-1}$; pink dots), carbon (mg l$^{-1}$; purple dots), and chlorophyll-α concentration (µg l$^{-1}$, green dots) of
SPOM collected during the both deployments by the McLane pump.
**Figure 9** Depth range of cold-water coral mound occurrences (blue shaded areas) at the (a) Namibian and (b) Angolan margins
in relation to the dissolved oxygen concentrations and potential temperature. Diurnal tides are delivering mainly phytodetritus
(shown in (a) and organic matter from the benthic nepheloid layer (shown in (b) as well as oxygen from above, and from below
to the mound sites (indicated by blue arrows, the length of which indicate the tidal ranges).
# 12. Tables
**Table 1.** Metadata of lander deployments conducted during RV *Meteor* cruise M122 (ANNA) in January 2016. The deployment
sites are shown in Figure 1.

|  | Station no. (GeoB ID) | Area | Lander | Date | Latitude [S] | Longitude [E] | Depth [m] | Duration [days] | Devices |
|---|---|---|---|---|---|---|---|---|---|
| **Namibia** | 20507-1 | on-mound | ALBEX | 01.-09.01.16 | 20°44.03' | 12°49.23' | 210 | 7.8 | + particle pump |
|  | 20508-1 | off-mound | TROL | 01.-09.01.16 | 20°44.03' | 12°49.14' | 220 | 7.8 |  |
|  | 20506-1 | off-mound | SLM | 01.-16.01.16 | 20°43.93' | 12°49.11' | 231 | 12.5 |  |
| **Angola** | 20921-1 | off-mound | ALBEX | 20.-23.01.16 | 9°46.16' | 12°45.96' | 342 | 2.5 | + particle pump |
|  | 20940-1 | off-mound | ALBEX | 23.-26.01.16 | 9°43.84' | 12°42.15' | 532 | 2.6 | + particle pump |
|  | 20916-1 | off-mound | TROL | 19.-26.01.16 | 9°43.66' | 12°42.09' | 526 | 6.8 |  |
|  | 20915-2 | off-mound | SLM | 19.-26.01.16 | 9°43.87' | 12°43.87' | 430 | 6.8 |  |


**Table 2** Tidal analysis of the ALBEX lander from 6 m above the sea floor. Depth, mean current speed, polarization ratio, mean
current direction, tidal prediction of pressure fluctuations, two most important harmonics with amplitude, tidal prediction of
horizontal current field, two most important harmonics with semi-major axis' amplitude.

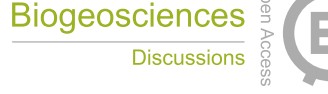

|  | Station no. (GeoB ID) | Depth (m) | Mean current speed (cm s⁻¹) | Polarization ratio | Current direction (°) | Tides [%] (p) | Const. [dbar] | Tides [%] (u) | Const. [cm s⁻¹] |
|---|---|---|---|---|---|---|---|---|---|
| **Namibia** | 20507-1 | 433 | 9.34 | 0.18 | 221.6 | 81.8 | M2: 0.37 | 10.5 | M2: 3.1 M3: 0.8 |
| **Angola** | 20921-1 | 532 | 9.96 | 0.42 | 247.9 | 91.6 | M2: 0.59 M3: 0.04 | 36 | M2: 7.8 M8: 0.7 |
|  | 20940-1 | 228 | 8.92 | 0.86 | 275.6 | 86.8 | M2: 0.60 M8: 0.02 | 50.9 | M2: 8.6 M3: 3.7 |

