# Peer review of "Environmental factors influencing benthic"

_Biogeosciences, 2019_

## Referee Comment (RC1) · Anonymous Referee #1 · 15 May 2019

The manuscript by Hanz et al. titled 'Environmental factors influencing cold-water coral ecosystems in the oxygen minimum zones on the Angolan and Namibian margins' report observations of live and extinct coral mounds and associated fauna along the southwestern margin of Africa (South Atlantic). The authors contrast 2 areas showing distinct cold-water corals patterns, one barren (Namibian margin) and one thriving (Angolan margin). The authors couple these observations with oceanographic properties in the vicinity of these mounds acquired with benthic landers and CTD. The authors report interesting findings: cold-water corals (and associated fauna) are not thought to occur at such low oxygen concentrations, and therefore is provided a detailed rationale of the various physical processes that could maintain the existence of these corals in

(perhaps short-lived) hypoxic conditions.

The manuscript is well written and provide interesting insights and details on the ecology and physiology of cold-water corals, here the scleractinian Lophelia pertusa. Given that observations of other megafauna are reported – e.g. along the Namibian margin on extinct coral mounds – this study is also broadly relevant to deep-sea biology, especially in the context of the presence of Oxygen Minimum Zones.

I enjoyed reading the manuscript and consider it an important contribution to the field of deep-sea biology. It is very relevant to obtain this ecological information to more accurately forecast impacts of a changing ocean and constrain habitat suitability models. I do not have major comments on the content of the manuscript. My comments are very specific (needed clarifications) and most relate to technical corrections.

Specific comments:

L291-293: The South Atlantic Subtropical Surface Water (SASSW) is not described in Section 2.1.1. Oceanographic setting. Could you add a short specification about the origin of this water mass to situate the reader?

Figure 4: Please specify geographic orientation relative to land (i.e. on the right?). I'm also confused by the statement at L309 that the OMZ was stretching at least 100 km offshore. Only 50 km is shown in the figure. Is this accurate or am I misunderstanding the figure?

Technical corrections:

L35: 'barotropic'

L104: What does the 7° refer to? Geographic coordinates, temperature? Please specify.

Figure 1: There is no a, b and c on the figure.

L211: No comma after 'Both'.

L226: water column in 2 words.

L242: The citation for R should read 'R Core Team, Year'.

L243: 'shorter term trends'

L281: "free waves"

L284: using

L323: Is the date accurate? Year is 2018?

L470: a temporal

L517: Did you mean Namibian margin?

L524: limits3

L577: no comma after both

L595: that

---

## Referee Comment (RC2) · Anonymous Referee #2 · 24 Jul 2019

This paper, according to the title, is about environmental factors influencing cold-water coral ecosystems in the oxygen minimum zones on the Angolan and Namibian margins. It describes results from a cruise off the southwest coast of the continent of Africa carried out in 2016. Specifically the cruise targeted two areas where previous work has suggested the presence of cold-water coral reef structures, one off the coast of Namibia and one further north off the coast of Angola. At each of the 2 sites landers were deployed, and across each site transects of CTD casts were made. There are numerous problems with the manuscript as it is written, many of which could easily be rectified. The first is that the paper is not really about cold-water coral ecosystems. The structures sampled in the Namibian sector are home to a deep water assemblage

which just happens to have grown on the relict remains of a cold-water reef that died thousands of years ago. It could just have easily grown on emerging bed-rock, an oil platform or a relatively recent wreck. While what was found may be informative about hard-substrate dwelling assemblages, it tells us nothing about cold-water corals. The findings from the Angolan sector, on the other hand, do contain information relevant to our knowledge about cold-water corals, specifically extending our knowledge about the environmental envelope within which reefs may be able to survive, with some limited evidence for mechanisms which may assist their survival. Overall the question that the manuscript raises is why there are no living corals off Namibia, given that the conditions off Angola are not highly different and all the factors that apparently mitigate low oxygen there, such as abundant high quality food and tidal excursions replenishing depleted oxygen, are also present off Namibia where they support a different assemblage. Much of the manuscript is unfocussed and over-detailed. It reads like a cruise report. A large proportion of the information given is presented in formats that are difficult to digest and are ultimately irrelevant in the context of the paper. We do not need to know that 2 landers were deployed but only the data from one is used here, for example. Samples for particulates and photo-pigments may have been collected from the CTDs, but there is no evidence analyses of these samples are used in the manuscript. And so on. Several pages of text could be replaced by a table and/or as supplementary material, allowing the reader to focus on the portions of the data and interpretation that are directly relevant to the subject of the paper, namely factors influencing deep-water hard-substratum assemblages and supporting their survival in zones of reduced oxygen availability.

Specific comments: P2 L35 Barotropic not barotrophic P2 L37-39 Dead coral mounds are not CWCs, so a complete rewrite with consistent nomenclature is recommended P2 L45 'Compensate' should be followed by 'for' P3 L54 et seq. Spacing among references is needed P3 L72 If aragonite saturation is important why is it not mentioned in the rest of the manuscript, and why was it not measured in this study? P3 L77 If a specific density envelope is important why is it not mentioned in the rest of the manuscript,

and why was it not measured? P5 L114-118 Are key parameters influencing CWC growth and therefore mound development really the focus of this investigation? What do the surveys from Namibia tell us about CWC growth? There are no living CWCs there. What are the new insights into susceptibility? P5 L127 All acronyms (here OMZ) should be defined on first occurrence in the manuscript. P6 L152-P7 L166 There were no CWCs at the Namibian site, only dead rubble with limited deep-water hard-bottom assemblages. P8 L180-190 Important records and details of the biological communities were recorded, begging the question why more was not made of this data in the paper. Many fish species were recorded in the Angolan reefs, which presumably aren't all OMZ specialists. P9 et seq. Methods and results. We do not need complete details of everything that was done on the cruise, the cruise report is already referenced, we only need the sampling and analysis details for the variables of relevance to this paper. Much could be done to condense text into a table or SI, to considerably shorten and focus the manuscript. P10 L240 Why was turbidity data only collected from Angola, and were the data used? P10 L274 What instrument was used to analyse the absorbtion spectrum etc? P11 L284-285 'unsinf' - ? Why were the data mean and trends removed? P12 L294 Why was 'SASSW' not discussed in the section describing water masses earlier? SCAW should presumably be SACW. The definition of SACW belongs in that section, not in the results. P12 L297 Temperature differences must not be confused with actual temperatures. The -1.3 and -0.2 here are differences but they are reported as a values. This is a problem throughout the manuscript. AAIW was not mentioned in the section on water masses, and should have been. P12 L305 et seq. Why DOconc and not simply DO, or even DO2? Abbreviations should be defined on first use. P13 L321 Table 1 is only metadata. A table of actual data would be helpful and reduce the need for a lot of text. P13 L324 et seq. Are the r values Spearman rank correlations? What is the justification for this approach? Are values truly independent? Would a more multivariate approach not have been more appropriate? Why do correlations between temperature and DO switch from negative to positive? P14 L331 It is unclear to this non-expert what several of the variables in Table 2 actually are/mean.

P14 L333 'whereafter', not 'where after' P14 L327-341 Does this section not simply describe what is well known about the forces (e.g. along-shore winds) driving upwelling along this coast? Why is what is known not reviewed or discussed in more detail in this manuscript? P14 L342-347 Isn't this the key (and only really relevant) result? More should be made of it. Is the method appropriate for calculating such incursions? P16 L356-372 Was CTD data not used? P16 L374 The figure encapsulates all that we need to know about POM inputs, so the text should describe what it shows and the authors are encouraged to leave out much of the irrelevant details elsewhere in the manuscript. The figure also combines details from both sites. The authors could shorten the manuscript by producing combined sections comparing and contrasting the sites, rather than describing the 2 sites separately. P16 L386 What is this surface water layer – a river plume? Should it not be described in the section about hydrography? P18 L408 Should this not read 'from the shallowest to the deepest'? Some of this section is confusing. P19 L425 What does 'p<0.01, deep' mean? P19 L427-430 This is the result of relevance and should be focused on. P21 L438 Would lower TOC and N in deeper waters not be expected? Could some of this text not be replaced with a figure, or is it repeating what is already in the figure? P21 L454 I do not accept that what was observed off Namibia can be regarded as a CWC. P21 L461 Who says seasonality has a major impact? Reference(s)? A better review and incorporatin of what is known about this coast needs to be included in the manuscript. P21 L465 If the measurements made in this study are not the relevant ones, what is the point of the whole manuscript? P22 L468 What is ESACW? This wasn't mentioned before. P22 L470 'a temporal' not 'an temporal' P22 L472-477 References are needed for all the statements in this paragraph (and espewhere in the manuscript). P22 L484 Some of this paragraph belongs in the results. P22 L495 How can the authors, based on limited cruise data, possibly determine what determines the absence of living CWCs from the Namibian margin? P23 L517 'Namibian' not 'Angolan'? P23 L518 DO is an input to habitat suitability modelling, not an output, surely? P23 L524 Not 'limits3' P24 L530-539 The conclusion appears to be drawn that increased food availability compensates

for decreased oxygen or higher temperature. Is it not the case that increased food in the water column is actually one of the main causes of decreased oxygen availability in these regions? This doesn't seem to be mentioned anywhere. P24 L551 What does 'loss of energy which and associated increased energy demand like' mean? P24 L555 'an' before 'energy (food) availability' unnecessary. P25 L563 If high quality food is available off Namibia but there are no living corals how can it be concluded that the presence of the SPOM promotes and/or supports coral growth? P25 L579 'leading to' not 'leads to' P25 L581-584 Some of this information belongs in the results section. P25 L585 Why does terrestrial POM constitute a less suitable food source, and who says so (references)? P25 L589 Delivery rates of SPOM were not measured, only the presence of POM with speculation as to its source(s). P26 L592 What is the source of this fresh POM? P26 L595 'fact that' not 'fact, that's' P26 L603 How are these currents likely to be responsible for the delivery of fresh SPOM from the surface productive zone? P26 L610 I do not understand how the nepheloid layer is formed by bottom erosion due to the intensification of near-bottom water movements, which is indicated by maxima of the buoyancy frequency N2 in 225 and 300 m depth. Explain and provide evidence. P27 L622 et seq. The examples of ecological roles of CWCs are not applicable in OMZs. P27 L626-627 CWCs are sometimes able to cope with low oxygen levels (there are none off Namibia).

---

## Author Response (AR1)

First of all we would like to thank the reviewer for the positive and helpful comments. We carefully went through all the comments and suggestions. We have adjusted the manuscript according to the comments made. Below we provide a description of the adjustments made, addressing the reviewers remarks.

Kind regards,

Ulrike Hanz (corresponding author)

*Anonymous Referee #1*

*The manuscript by Hanz et al. titled 'Environmental factors influencing cold-water coral ecosystems in the oxygen minimum zones on the Angolan and Namibian margins' report observations of live and extinct coral mounds and associated fauna along the southwestern margin of Africa (South Atlantic). The authors contrast 2 areas showing distinct cold-water corals patterns, one barren (Namibian margin) and one thriving (Angolan margin). The authors couple these observations with oceanographic properties in the vicinity of these mounds acquired with benthic landers and CTD. The authors report interesting findings: cold-water corals (and associated fauna) are not thought to occur at such low oxygen concentrations, and therefore is provided a detailed rationale of the various physical processes that could maintain the existence of these corals in (perhaps short-lived) hypoxic conditions. The manuscript is well written and provide interesting insights and details on the ecology and physiology of cold-water corals, here the scleractinian Lophelia pertusa. Given that observations of other megafauna are reported – e.g. along the Namibian margin on extinct coral mounds – this study is also broadly relevant to deep-sea biology, especially in the context of the presence of Oxygen Minimum Zones.*
*I enjoyed reading the manuscript and consider it an important contribution to the field of deep-sea biology. It is very relevant to obtain this ecological information to more accurately forecast impacts of a changing ocean and constrain habitat suitability models.*
*I do not have major comments on the content of the manuscript. My comments are very specific (needed clarifications) and most relate to technical corrections.*

**Reply on specific comments:**

L291-293: The South Atlantic Subtropical Surface Water (SASSW) is not described in Section 2.1.1. Oceanographic setting. Could you add a short specification about the origin of this water mass to situate the reader?

**A short description of SASSW was added.**

Figure 4: Please specify geographic orientation relative to land (i.e. on the right?). I'm also confused by the statement at L309 that the OMZ was stretching at least 100 km offshore. Only 50 km is shown in the figure. Is this accurate or am I misunderstanding the figure?

**This is a misunderstanding coming from the Figure. Figure 4 does not show the 50 km closest to the shore, but the shore is about 45 km further to the east (right).**

**Technical corrections:**

L35: 'barotropic'
**Changed.**

L104: What does the 7_ refer to? Geographic coordinates, temperature? Please specify.
**It refers to the geographical coordinate. We changed it to 7° S.**

Figure 1: There is no a, b and c on the figure.
**The figure was changed accordingly.**

L211: No comma after 'Both'.
**The comma was removed.**

L226: water column in 2 words.
**Changed.**

L242: The citation for R should read 'R Core Team, Year'.
**The citation was updated.**

L243: 'shorter term trends'
**Changed.**

L281: "free waves"
**Changed.**

L284: using
**Changed.**

L323: Is the date accurate? Year is 2018?
**Indeed this is not accurate. It should be 2016. We modified the text.**

L470: a temporal
**Changed.**

L517: Did you mean Namibian margin?
**Yes we did. The text was changed accordingly.**

L524: limits3
**Changed.**

L577: no comma after both
**The comma was removed.**

L595: that
**Changed.**

We wish to thank the reviewer for the efforts and input provided. We carefully went through all the comments and suggestions and have adjusted the manuscript accordingly. Below we provide a description of the adjustments made, addressing the reviewers remarks.

Kind regards,

Ulrike Hanz (corresponding author)

**Anonymous Referee #2**

This paper, according to the title, is about environmental factors influencing cold-water coral ecosystems in the oxygen minimum zones on the Angolan and Namibian margins. It describes results from a cruise off the southwest coast of the continent of Africa carried out in 2016. Specifically the cruise targeted two areas where previous work has suggested the presence of cold-water coral reef structures, one off the coast of Namibia and one further north off the coast of Angola. At each of the 2 sites landers were deployed, and across each site transects of CTD casts were made. There are numerous problems with the manuscript as it is written, many of which could easily be rectified. The first is that the paper is not really about cold-water coral ecosystems. The structures sampled in the Namibian sector are home to a deep water assemblage which just happens to have grown on the relict remains of a cold-water reef that died thousands of years ago. It could just have easily grown on emerging bed-rock, an oil platform or a relatively recent wreck. While what was found may be informative about hard-substrate dwelling assemblages, it tells us nothing about cold-water corals.

**We agree that the ecosystem on the Namibian margin is not a cold-water coral ecosystem. We have emphasized in the text that it is a deep-water assemblage of sponges and bryozoans that is growing on cold-water coral remains, which grows in extremely low oxygen concentrations. We also have adapted the title of the manuscript.**

The findings from the Angolan sector, on the other hand, do contain information relevant to our knowledge about cold-water corals, specifically extending our knowledge about the environmental envelope within which reefs may be able to survive, with some limited evidence for mechanisms which may assist their survival. Overall the question that the manuscript raises is why there are no living corals off Namibia, given that the conditions off Angola are not highly different and all the factors that apparently mitigate low oxygen there, such as abundant high quality food and tidal excursions replenishing depleted oxygen, are also present off Namibia where they support a different assemblage. Much of the manuscript is unfocussed and over-detailed. It reads like a cruise report. A large proportion of the information given is presented in formats that are difficult to digest and are ultimately irrelevant in the context of the paper. We do not need to know that 2 landers were deployed but only the data from one is used here, for example.

**We have shortened the Materials and Method section and went through the manuscript to remove unnecessary information.**

Samples for particulates and photo-pigments may have been collected from the CTDs, but there is no evidence analyses of these samples are used in the manuscript.

**We have removed this from the manuscript, see comment above.**

And so on. Several pages of text could be replaced by a table and/or as supplementary material, allowing the reader to focus on the portions of the data and interpretation that are directly relevant to the subject of the paper, namely factors influencing deepwater hard-substratum assemblages and supporting their survival in zones of reduced oxygen availability.

Specific comments:

P2 L35 Barotropic not barotrophic
**Changed.**

P2 L37-39 Dead coral mounds are not CWCs, so a complete rewrite with consistent nomenclature is recommended
**It was changed in all relevant sections of the manuscript.**

P2 L45 'Compensate' should be followed by 'for'
**We have added "for".**

P3 L54 et seq. Spacing among references is needed
**A spacing was added to all relevant references.**

P3 L72 If aragonite saturation is important why is it not mentioned in the rest of the manuscript, and why was it not measured in this study?
**In this study we did not focus on the aragonite saturation even though it is an important factor. It is expected to not be a limiting factor in the Atlantic at these depths.**

P3 L77 If a specific density envelope is important why is it not mentioned in the rest of the manuscript, and why was it not measured?
**The appearance of CWCs in relation to the density envelope has been added to the discussion (L653ff).**

P5 L114-118 Are key parameters influencing CWC growth and therefore mound development really the focus of this investigation? What do the surveys from Namibia tell us about CWC growth? There are no living CWCs there. What are the new insights into susceptibility?
**The focus of this manuscript is on benthic communities growing on coral mounds in oxygen depleted environments. We agree with the reviewer that the communities on the Namibian margin are no CWC ecosystems. We adapted the manuscript accordingly (see comment above).**

P5 L127 All acronyms (here OMZ) should be defined on first occurrence in the manuscript.
**OMZ was defined in L117.**

P6 L152-P7 L166 There were no CWCs at the Namibian site, only dead rubble with limited deep-water hard-bottom assemblages.
**We have changed the description of the benthic community to avoid confusion.**

P8 L180-190 Important records and details of the biological communities were recorded, begging the question why more was not made of this data in the paper. Many fish species were recorded in the Angolan reefs, which presumably aren't all OMZ specialists.
**This was unfortunately not the focus of this manuscript, whereas we do agree that this information is very interesting. These data will be part of other manuscripts.**

P9 et seq. Methods and results. We do not need complete details of everything that was done on the cruise, the cruise report is already referenced, we only need the sampling and analysis details for the variables of relevance to this paper. Much could be done to condense text into a table or SI, to considerably shorten and focus the manuscript.
**We do agree with the reviewer. We have shortened the text to only focus on relevant information.**

P10 L240 Why was turbidity data only collected from Angola, and were the data used?
**Unfortunately no turbidity data were collected on the Namibian margin, due to technical issues with the sensor on the CTD. The data is shown in Figure 5 (CTD transect across the Angolan margin).**

P10 L274 What instrument was used to analyse the absorbtion spectrum etc?
**A Waters Acquity UPLC system was used (was added to material and methods L300)**

P11 L284-285 'unsinf' - ? Why were the data mean and trends removed?
**The tidal analysis outcome will not change in respect of significant constituents or their amplitudes whether or not the means and trends are within the input data. It belongs to the standard procedure of tidal analysis.**

P12 L294 Why was 'SASSW' not discussed in the section describing water masses earlier? SCAW should presumably be SACW. The definition of SACW belongs in that section, not in the results.
**A short description of SASSW and AAIW was added (L 160ff) and the definition of SACW was moved to section 2.1.1.**

P12 L297 Temperature differences must not be confused with actual temperatures. The -1.3 and -0.2 here are differences but they are reported as a values. This is a problem throughout the manuscript. AAIW was not mentioned in the section on water masses, and should have been.
**We agree that this might be confusing. Temperature differences are now reported as Δ values. AAIW is now mentioned in section 2.1.1. (L164).**

P12 L305 et seq. Why DOconc and not simply DO, or even DO2? Abbreviations should be defined on first use.
**$DO_{conc}$ is the shortest abbreviation for dissolved oxygen (DO) concentration. The definition was added.**

P13 L321 Table 1 is only metadata. A table of actual data would be helpful and reduce the need for a lot of text.

**An additional table with values from the lander deployments at the Angolan and Namibian margin was added (Table 2).**

P13 L324 et seq. Are the r values Spearman rank correlations? What is the justification for this approach? Are values truly independent? Would a more multivariate approach not have been more appropriate? Why do correlations between temperature and DO switch from negative to positive?
**Spearman rank correlations were used because they show a general statistical correlation of two variables (water characteristics). Oxygen concentrations for example are not independent from temperature, whereas these dependencies are not strongly influencing the outcome since we were not investigating underlying causes for the correlations, but wanted to show how the separate variables correlate.**
**Correlations did not switch from negative to positive. Both datasets from Angola come from two separate deployments in two separate depths and therefore show to separate correlations. We have indicated this more clearly in Fig 7.**

P14 L331 It is unclear to this non-expert what several of the variables in Table 2 actually are/mean.
**I removed the polarization ratio. Other variables are explained in the table caption.**

P14 L333 'whereafter', not 'where after'
**Changed.**

P14 L327-341 Does this section not simply describe what is well known about the forces (e.g. along-shore winds) driving upwelling along this coast? Why is what is known not reviewed or discussed in more detail in this manuscript?
**Yes, it serves as an example how the seasonal variations change the water characteristics at the margin. It is reviewed in section 4.1.**

P14 L342-347 Isn't this the key (and only really relevant) result? More should be made of it. Is the method appropriate for calculating such incursions?
**It would of course be much nicer to be able to capture these incursions with for example a mooring comprising the whole water column, but unfortunately we were not able to use a mooring in this study. We think that this method is the best method we can use with the available data. We are aware that this is an estimation.**

P16 L356-372 Was CTD data not used?
**Not used in this manuscript. (Changed in the method section, L275ff).**

P16 L374 The figure encapsulates all that we need to know about POM inputs, so the text should describe what it shows and the authors are encouraged to leave out much of the irrelevant details elsewhere in the manuscript. The figure also combines details from both sites. The authors could shorten the manuscript by producing combined sections comparing and contrasting the sites, rather than describing the 2 sites separately.
**We have changed the result section according to the recommendation made and combined both sites.**

P16 L386 What is this surface water layer – a river plume? Should it not be described in the section about hydrography?
**The river influence is now described in section 2.1.1. (L164f).**

P18 L408 Should this not read 'from the shallowest to the deepest'? Some of this section is confusing.
**The paragraph was changed to remove eventual confusion.**

P19 L425 What does 'p<0.01, deep' mean?
**It means that the correlation of turbidity and oxygen concentration during the deep deployment is significant. We changed the description to make it more clear.**

P19 L427-430 This is the result of relevance and should be focused on.
**It is focused on in section 4.4 and Figure 9. We have tried to stress the importance slightly more by adapting the text in section 4.4.**

P21 L438 Would lower TOC and N in deeper waters not be expected? Could some of this text not be replaced with a figure, or is it repeating what is already in the figure?
**Yes, it is expected. It is a description of what is shown in Figure 8.**

P21 L454 I do not accept that what was observed off Namibia can be regarded as a CWC.
**We have made the changes throughout the whole manuscript, see also response above.**

P21 L461 Who says seasonality has a major impact? Reference(s)? A better review and incorporation of what is known about this coast needs to be included in the manuscript.
**Relevant references were added to section 4.1.**

P21 L465 If the measurements made in this study are not the relevant ones, what is the point of the whole manuscript?
**These are the first records of environmental factors in these two specific areas. These data provide valuable insight about daily environmental fluctuations even though it is a short term record. Main outcome is that the boundaries previously described on CWC oxygen tolerance need to be adapted. Indeed the data only provides a snapshot in time and ranges can be even larger. However, this does not devaluate the measurements. We still measured the lowest ever recorded oxygen concentrations for *L. pertusa*.**

P22 L468 What is ESACW? This wasn't mentioned before.
**ESACW was mentioned in L147.**

P22 L470 'a temporal' not 'an temporal'
**Changed.**

P22 L472-477 References are needed for all the statements in this paragraph (and elsewhere in the manuscript).
**References were added.**

P22 L484 Some of this paragraph belongs in the results.
**Specific values were removed.**

P22 L495 How can the authors, based on limited cruise data, possibly determine what determines the absence of living CWCs from the Namibian margin?
**We can not determine what determines the absence of living CWCs but we can hypothesize, since it is accepted that environmental conditions changed at the same time with the CWCs disappearance (Tamborrino et al. accepted).**

P23 L517 'Namibian' not 'Angolan'?
**Yes, Corrected.**

P23 L518 DO is an input to habitat suitability modelling, not an output, surely?
**It is only an input. Corrected.**

P23 L524 Not 'limits3'
**Corrected.**

P24 L530-539 The conclusion appears to be drawn that increased food availability compensates for decreased oxygen or higher temperature. Is it not the case that increased food in the water column is actually one of the main causes of decreased oxygen availability in these regions? This doesn't seem to be mentioned anywhere.
**We agree with the reviewer and have adapted the text accordingly (L661ff).**

P24 L551 What does 'loss of energy which and associated increased energy demand like' mean?
**It should be 'loss of energy with an associated increased energy demand'. Corrected.**

P24 L555 'an' before 'energy (food) availability' unnecessary.
**'An' was removed.**

P25 L563 If high quality food is available off Namibia but there are no living corals how can it be concluded that the presence of the SPOM promotes and/or supports coral growth?
**It can not be concluded for CWCs at the Namibian margin. It can only be suggested for the benthic fauna which is associated to the dead cold-water coral framework, since it still survives in otherwise stressful conditions.**

P25 L579 'leading to' not 'leads to'
**Corrected.**

P25 L581-584 Some of this information belongs in the results section.
**Information was removed from this section.**

P25 L585 Why does terrestrial POM constitute a less suitable food source, and who says so (references)?
**Because terrestrial matter includes carbon rich polymeric material (cellulose, hemicellulose and lignin) which cannot easily be taken up by marine organisms (Hedges and Oades, 1997). A reference and explanation was added (L605ff).**

P25 L589 Delivery rates of SPOM were not measured, only the presence of POM with speculation as to its source(s).
**True, the presence of SPOM was meant, corrected.**

P26 L592 What is the source of this fresh POM?
**It is directly sinking as well as advected organic matter from the surface ocean. Explanation added (L723f).**

P26 L595 'fact that' not 'fact, that's'
**Corrected.**

P26 L603 How are these currents likely to be responsible for the delivery of fresh SPOM from the surface productive zone?

**Currents lead to mixing of the water layers, which allows material from the surface water layer to sink down to the mound sites. Explanation was added to the manuscript (736f)**

P26 L610 I do not understand how the nepheloid layer is formed by bottom erosion due to the intensification of near-bottom water movements, which is indicated by maxima of the buoyancy frequency N2 in 225 and 300 m depth. Explain and provide evidence.

**The interaction of internal waves with the margin topography will intensify currents and therefore mixing. These internal waves can move on density gradients which are indicated by buoyancy frequency maxima. The manuscript was changed and a better explanation was added.**

P27 L622 et seq. The examples of ecological roles of CWCs are not applicable in OMZs.

**The benthic communities on the cold-water coral rubble were meant. We have changed the text.**

P27 L626-627 CWCs are sometimes able to cope with low oxygen levels (there are none off Namibia).

**This is due to the fact that oxygen levels are too low at the Namibian margin.**

**The line numbers refers to the track changes document.**

**List of relevant changes**

Spacing was added between all relevant references through the whole manuscript. Figures were moved and numbers were changed in all relevant cases.

P1 L2f Title was adapted

P2 L24f Abstract was adapted in order to stress that mounds on the Namibian margin do not include living CWCs. Non relevant information was removed from the abstract (L40f).

P2 L39 "Barotropic" was corrected.

P2 L52 "Compensate for" was corrected.

P5 L115: Geographical location (° S) was added to make clear that it is a geographical coordinate.

P6 L126 The absence of CWC at the Namibian margin was stressed.

P6 L150f The definition of SACW was moved to this section.

P7 L160ff A description of the water masses (SASSW and AAIW) and the influence of major rivers at the Angolan margin was added.

P8 Figure 1 a), b) and c) was added to the Figure.

P9 L182 CWC were changed to "community" to stress the absence of alive CWC.

P9 L187 It was stressed that only dead CWC were found at the Namibian margin.

P11 L221ff Non relevant information about the lander deployments was removed.

P11 L240 Text was changed in order to remove unnecessary information.

P11 L255 "Water column" was corrected.

P12 L255ff Non relevant information about the CTD transects was removed.

P12 L261 Explanation why turbidity was not measured at the Namibian margin was added.

P12 L270f Citation was corrected.

P12 L272 "Short term trends" was corrected.

P12 L275ff Non relevant information about the SPOM collection was removed.

P13 L299f Instrument name was added.

P14 L310 "Free waves" was corrected.

P14 L313 "Using" was corrected.

P14 L316ff Result section was changed according to the suggestion of referee 2. Environmental characteristics of the Namibian and Angolan margin are now presented consecutively.

P14 L323f Definition of SACW was moved to section 2.1.1. "SACW" was corrected.

P14 L327 Differences are now marked as delta values.

P15 L335 Abbreviation of DO was added.

P15 L339 and L347: Position of transect in regard to the coastline was described better.

P16 L352: CTD transects of the Angolan margin are now described before the results of other measurements.

P17 Figure 5 is now the CTD transect of the Angolan margin.

P17 L376ff Near bottom environmental data of both regions are now described consecutively.

P17 L380 Differences are now marked as delta values.

P18 L381 Date was corrected.

P18 L387ff Sentences were moved to stress the influence of wind on water characteristics on the Namibian margin.

P18 L397 "after" was corrected

P18 L407 The inaccuracy of the method was stressed.

P18 L411 Delta was added.

P19 L413 Number of figure was changed.

P20 L420 Near bottom environmental data of Angola was moved.

P21 L442 Figure was moved.

P22 L467 Paragraph was slightly changed to avoid confusion about the shallow and deep deployment.

P23 L480 Figure number was changed.

P27 L561ff Text was changed in order to stress that there are no CWC on the Namibian margin.

P27f References were added to section 4.1.

P28 L582 "A temporal" was corrected.

P28 L598ff Specific values were removed from the discussion.

P28 L605ff Description of the influence of terrestrial organic matter was added.

P29 L636 Namibian margin was added.

P30 L639 Oxygen concentration as a model output was removed.

P30 L643 "Limits" was corrected.

P30 L653 Density envelope was added to the discussion.

P30 L661ff Explanation of a negative feedback of high food availability on the oxygen concentration was added.

P31 L676ff Sentence was removed.

P31 L681 "An" removed.

P31f L688ff Text was shortened and specific values were removed from the discussion.

P32 L720 Delivery was replaced by availability.

P32 L724 Unnecessary text was removed.

P33 L736 Short description of how currents are responsible to mix the water masses and therefore are responsible in delivering SPOM to the mound areas.

P33 L743ff Section was changed to explain the function and location of the internal waves.

P34 L758ff Implications were fused with the conclusion.

P34 L769ff Text was changed to stress the absence of living CWCs on the Namibian margin.

P41 L1047 Reference was updated.

P44f L1159 Table with environmental properties of the Namibian and Angolan margin was added to the manuscript.

P45 L1163 Polarization ratio was removed from table 3.

**Environmental factors influencing  benthic communities in the oxygen minimum zones on the Angolan and Namibian margins**

Ulrike Hanz[1], Claudia Wienberg[2], Dierk Hebbeln[2], Gerard Duineveld[1], Marc Lavaleye[1], Katriina Juva[3], Wolf-Christian Dullo[3], André Freiwald[4], Leonardo Tamborrino[2], Gert-Jan Reichart[1,5], Sascha Flögel[3], Furu Mienis[1]

[1]*NIOZ-Royal Netherlands Institute for Sea Research and Utrecht University, Department of Ocean Systems, Texel, 1797SH, Netherlands*

[2]*MARUM–Center for Marine Environmental Sciences, University of Bremen, Bremen, 28359, Germany*

[3]*GEOMAR Helmholtz Centre for Ocean Research, Kiel, 24148, Germany*

[4]*Department for Marine Research, Senckenberg Institute, Wilhelmshaven, 26382, Germany*

[5]*Faculty of Geosciences, Earth Sciences Department, Utrecht University, Utrecht, 3512JE, Netherlands*

*Correspondence to*: Ulrike Hanz (ulrike.hanz@nioz.nl), +31222369466

**Abstract**

Thriving benthic communities were observed in the oxygen minimum zones along the southwestern African margin. On the Namibian margin  fossil cold-water coral mounds were overgrown by sponges and bryozoans, while  the Angolan margin was characterized by cold-water coral mounds covered by a living coral reef . ~~Fossil cold-water coral mounds overgrown by sponges and bryozoans were observed in anoxic conditions on the Namibian margin, while mounds colonized by thriving cold-water coral reefs were found in hypoxic conditions on the Angolan margin. These low oxygen conditions do not meet known environmental ranges favoring cold-water corals and hence are expected to provide unsuitable habitats for cold-water coral growth and therefore reef formation.the livingfaunacan nevertheless thriveat the southwestern African margin. Downslopethe deployment ofwere usedTemporalin the mound areasoscillating17on the Namibianon the Angolan marginhwith excursions of up to 70 and 130 m for the Namibian and Angolan margins, respectivelytemporarilycoral moundshypoxicof suspended particulatewhich serves as acold-water coralsslopeindicates a completely marine source ands, whereas onmaterialsssmound area on the Angolan marginsfreshin both areasThissthatas well as the associated faunainduced byan.OWith the expected expansion of oxygen~~

[revised manuscript text omitted]

---

## Author Response (AR2)

**First of all we would like to thank the reviewer for the positive feedback and the helpful comments. We carefully went through all the comments and suggestions. We have adjusted the manuscript according to the comments made. Below we provide a description of the adjustments made, addressing the reviewers remarks.**

**Kind regards,**

**Ulrike Hanz (corresponding author)**

This is a greatly improved manuscript, and the authors appear to have addressed most of the concerns raised in this reviewer's previous assessment. The manuscript would be further improved by addressing some minor issues, as follows:

L26 The study does not really 'explain' why benthic communities differ, so perhaps 'explore' would be a better word to use.
- **'Explain' was exchanged by 'explore'.**

L81 replace ( with ; before Fink
- **Was replaced.**

L107 insert hyphen between reef and covered
- **Hyphen was added.**

L109 The study does not really 'identify' why CWCs thrive, so perhaps 'explore' would be a better word to use.
- **'Identify' was replaced by 'explore'.**

L112 The data are not really 'used to provide new insights in susceptibility'. Replace last sentence with 'The data are used to improve understanding of the potential fate of CWC mounds in a changing ocean'.
- **Sentence was replaced.**

L124 OMZ should be singular in this sentence (delete s)
- **S was deleted.**

L141 replace last part of sentence to read 'expanding from the oceanic zone 350 km offshore towards the coast'
- **Last part of the sentence was replaced.**

L143 insert hyphen between sun and warmed
- **Hyphen was added.**

L150 The TROL lander was not used in this study, so remove from the figure (and text)

- **TROL lander was removed.**

L166 change 'underneath' to 'in'
- **Preposition was changed.**

L170 move 'also' to between 'were' and 'present' on L171
- **'Also' was moved.**

L172 insert 'along with some' before M. oculata.
- **'Along with some' was inserted.**

L174 replace 'an even' with 'a'
- **Was replaced.**

L185 change to 'hide in hollows in the coral framework'
- **Was changed.**

L199 delete sentence about TROL
- **Sentence was deleted.**

L218 Change to 'Owing to technical problems turbidity data were only collected'
- **Was changed.**

L226 Question: what is the rationale for using Spearman rank correlations, and how do you deal with non-independence among the observations? [This is not a show stopper, but I am still wondering].
- **Alternatively we could have also used Pearson's correlation, which is the more simple solution, whereas this would limit the correlations to linear relationships. Spearman rank correlations gives an indication about the strength and the direction of the correlation between two variables. We are just interested in showing general trends (positive or negative) instead of precise mathematical correlations which can be used to predict factors. We just want to see if maxima and minima are aligning or not. Therefore the non-independence is not a problem. Oxygen for example depends on temperature but also on other factors. The correlation between them is nevertheless different in the shallow and deeper coral mound areas since their influence on each other is not big enough to mask the different water mass movements.**

L236 commas after filters and analysis are unnecessary – delete
- **Commas were removed.**

L240 the 'used' is unnecessary – delete
- **'Used' was deleted.**

L241 'standards' not 'a standard', and 'were' not 'was'
- **Changed.**

L242 'exercised' is not the right word – use 'used'
- **Changed.**

L244 'comparison with' not 'comparing'
- **Replaced.**

L254 replace 'about' with 'of'
- **Replaced.**

L281 et seq. There is a real problem with using DOconc to mean DO concentration. Why can you not use simply DO, or better, DO concentration? If you use the latter you can simply distinguish between DO concentration (singular) and DO concentrations (plural). As DOconc is used it is often impossible to determine which is meant. Maybe use DOconc and DOconcs?
- **We agree on the singular-plural problem. We changed it to DO concentration(s).**

L284 replace 'was stretching' with 'stretched'
- **Replaced.**

L285 replace 'towards' with 'to'
- **Replaced.**

L291 delete 'up to'
- **Deleted.**

L293 the meaning of 'towards 50 km from the coastline' is very unclear. Why is the 'section distance' increasing from left to right, assuming that the coast is towards the right of the plots? Why not replace the x axis with 'distance to the coast'?
- **The section distance is a section on the continental margin which is not directly connected to the coastline. The section distance gives us an estimate about the size of the transect which is also shown in Figure 1.**

L308 should read 'decreased further to a minimum of' … 'and then increased to'
- **Changed.**

L311-2 is overall DOconc singular or plural in this sentence? I think singular, in which case it should be 'was' on L312
- **Changed to singular.**

L317 again the x axes are confusing. Replace with 'distance to the coast'?
- **See explanation above.**

L329 not clear what 'especially' is meant to mean – replace with 'markedly'?

- **Was replaced.**

L340 change to 'changed to a dominantly northerly direction'

- **Changed.**

L356 temperatures were (not was)

- **Changed.**

L358 DOconc here is plural, so change it (see note above)

- **$DO_{conc}$ was changed, like suggested above.**

L362 delete 'overall'

- **Deleted.**

L366 delete comma after 'showed'

- **Comma was deleted.**

L396 change to 'Carotenoids (0.08-0.12 µg l-1) and fucoxanthin (0.22 µg l-1), which are common in diatoms, were major components of the pigment fraction.'

- **Sentence was changed.**

L400 insert 'were' after SPOM

- **'Were' was inserted.**

L408 delete 'an', change 'ratio' to 'ratios' and replace 'on' with 'at'

- **Was changed.**

L426 et seq. 'likely' on its own as an adverb is not good English. Use alternatives such as 'probably', 'are likely to', 'may', 'might' and so on. The sentence could read 'It is probably that differences in present-day environmental conditions between the areas influence the faunal assemblages inhabiting them'.

- **'Likely' was changed in all relevant cases.**

L432 lowest DOconc is singular so 'is' not 'are'

- **Changed.**

L434 replace 'most likely' with 'probably'

- **Likely was replaced.**

L440 replace 'was leading' with 'led'

- **Changed.**

L446 replace 'likely' with 'possibly'
- **Likely was replaced.**

L468 DOconc is plural, so change to reflect this
- **Changed.**

L494 delete 'low' at the end of the line
- **Deleted.**

L520 'stresses' not 'stress'
- **Changed.**

L524 insert comma after CWCs
- **Inserted.**

L545 'coupled to periods of other environmental stressors' makes no sense. Do you mean 'Variations in food quality…, which were relatively small during this study, did not seem to be related to the presence of other environmental stressors'?
- **Yes, indeed that is what we mean. Was changed to the suggested solution.**

L549 replace 'likely' with 'probably'
- **Likely was replaced.**

L552 replace 'likely' with 'probably'
- **Likely was replaced.**

L555 replace 'likely' with 'possibly'
- **Likely was replaced.**

L562 change to 'may be unsuitable for CWCs'
- **Changed.**

L580 replace 'likely defines the presence of CWCs' with 'probably defines the limits of suitable habitat for CWCs'
- **Sentence was replaced.**

L581 'stress' not 'stressor'
- **Changed.**

L594 replace 'is accumulated' with 'accumulates'
- **Replaced.**

**List of relevant changes**

L26 'Explain' was exchanged by 'explore'.
L81 '(' was replaced by '; '.
L107 Hyphen was added.
L109 'Identify' was replaced by 'explore'.
L112 Sentence was replaced by 'The data are used to improve understanding of the potential fate of CWC mounds in a changing ocean'.
L124 'S' was deleted.
L141 Last part of the sentence was replaced by 'expanding from the oceanic zone 350 km offshore towards the coast'.
L143 Hyphen was added.
L150 TROL lander was removed from figure.
L166 Preposition was changed.
L170 'Also' was moved.
L172 'Along with some' was inserted.
L174 'An even' was replaced with 'a'.
L185 Was changed to 'hide in hollows in the coral framework'.
L199 Sentence about TROL was deleted.
L218 Was changed to 'Owing to technical problems turbidity data were only collected'.
L236 Commas were removed.
L240 'Used' was deleted.
L241 'A standard' was replaced by 'standards' and 'was' was replaced by 'were'.
L242 'Exercised' was replaced by 'used'.
L244 'Comparing' was replaced by 'comparison with'.
L254 'About' was replaced with 'of'.
L281 $DO_{conc}$ was replaced by DO concentration(s) throughout the manuscript.
L284 'Was stretching' was replaced with 'stretched'.
L285 'Towards' was replaced with 'to'.
L291 'Up to' was deleted.
L308 Changed to 'decreased further to a minimum of' … 'and then increased to'.
L311 Changed to 'was'.
L329 'Especially' was replaced by 'markedly'.
L340 Sentence was changed to 'changed to a dominantly northerly direction'.
L356 'Was' was changed to 'were'.
L358 $DO_{conc}$ was changed to DO concentrations.
L362 'Overall' was deleted.
L366 Comma was deleted.
L396 Sentence was changed to 'Carotenoids (0.08-0.12 µg l-1) and fucoxanthin (0.22 µg l-1), which are common in diatoms, were major components of the pigment fraction.'
L400 'Were' was inserted.
L408 'An' was deleted, 'ratio' was changed to 'ratios' and 'on' was replaced with 'at'.
L426 'Likely' was changed in all relevant cases.

L432 'Are' was changed to 'is'.

L434 'Most likely' was replaced with 'probably'.

L440 'Was leading' was replaced with 'led'.

L446 'Likely' was replaced with 'possibly'.

L468 $DO_{conc}$ was changed to DO concentrations.

L494 'Low' was deleted.

L520 'Stress' was changed to 'stresses'.

L524 Comma was inserted.

L545 Sentence was changed to 'Variations in food quality…, which were relatively small during this study, did not seem to be related to the presence of other environmental stressors'.

L549 'Likely' was replaced with 'probably'.

L552 'Likely' was replaced with 'probably'.

L555 'Likely' was replaced with 'possibly'.

L562 Sentence was changed to 'may be unsuitable for CWCs'.

L580 'Likely defines the presence of CWCs' was replaced with 'probably defines the limits of suitable habitat for CWCs'.

L581 'Stressor' was replaced by 'stress'.

L594 'Is accumulated' was replaced with 'accumulates'.

[revised manuscript text omitted]